# The SUMO–NIP45 pathway processes toxic DNA catenanes to prevent mitotic failure

Emil P. T. Hertz [1] ✉, Ignacio Alonso-de Vega[1], Thomas Kruse[1], Yiqing Wang [2,3], Ivo A. Hendriks [4], Anna H. Bizard [2,3], Ania Eugui-Anta[5], Ronald T. Hay [5], Michael L. Nielsen [4], Jakob Nilsson [1], Ian D. Hickson [2,3] & Niels Mailand [1,2] ✉

SUMOylation regulates numerous cellular processes, but what represents the essential functions of this protein modification remains unclear. To address this, we performed genome-scale CRISPR–Cas9-based screens, revealing that the BLM-TOP3A-RMI1-RMI2 (BTRR)-PICH pathway, which resolves ultrafine anaphase DNA bridges (UFBs) arising from catenated DNA structures, and the poorly characterized protein NIP45/NFATC2IP become indispensable for cell proliferation when SUMOylation is inhibited. We demonstrate that NIP45 and SUMOylation orchestrate an interphase pathway for converting DNA catenanes into double-strand breaks (DSBs) that activate the G2 DNA-damage checkpoint, thereby preventing cytokinesis failure and binucleation when BTRR-PICH-dependent UFB resolution is defective. NIP45 mediates this new TOP2-independent DNA catenane resolution process via its SUMO-like domains, promoting SUMOylation of specific factors including the SLX4 multi-nuclease complex, which contributes to catenane conversion into DSBs. Our findings establish that SUMOylation exerts its essential role in cell proliferation by enabling resolution of toxic DNA catenanes via nonepistatic NIP45- and BTRR-PICH-dependent pathways to prevent mitotic failure.

Protein modification by the polypeptide SUMO (small ubiquitin-like modifier) proceeds via a three-step enzymatic cascade and regulates numerous cellular processes including gene expression, cell-cycle progression and genome maintenance[1,2]. Most prominently, SUMOylation is a crucial mediator of many cellular stress responses. Accordingly, SUMO pathway components are frequently overexpressed in cancers to cope with high levels of stress correlating with poor prognosis[3]. Proteomic studies have shown that thousands of human proteins are modified by SUMOylation, and more than 40,000 individual SUMO acceptor sites in the human proteome have been identified[4,5]. Not surprisingly, therefore, SUMOylation is essential for viability in eukaryotic species

including yeasts, nematodes, flies and mammals[6–10]. In cultured human cells, inhibition of SUMOylation mainly subverts cell proliferation by undermining the fidelity of mitotic progression and chromosome segregation[11]. However, why SUMOylation is particularly critical for mitosis is unclear. More generally, whether the essential requirement of SUMO signaling for cell viability and proliferation reflects the composite effect of numerous individual SUMOylation events or entails specific SUMO-driven processes that are particularly critical for maintaining fitness is not known. The recent development of highly specific, small-molecule inhibitors of SUMOylation such as ML-792 (henceforth referred to as SUMOi) and its functional analog TAK-981, which potently

[1]Protein Signaling Program, Novo Nordisk Foundation Center for Protein Research, University of Copenhagen, Copenhagen, Denmark. [2]Center for Chromosome Stability, University of Copenhagen, Copenhagen, Denmark. [3]Center for Healthy Aging, Department of Cellular and Molecular Medicine, University of Copenhagen, Copenhagen, Denmark. [4]Proteomics Program, Novo Nordisk Foundation Center for Protein Research, University of Copenhagen, Copenhagen, Denmark. [5]Centre for Gene Regulation and Expression, School of Life Sciences, University of Dundee, Dundee, UK. ✉e-mail: emil.hertz@cpr.ku.dk; niels.mailand@cpr.ku.dk

inhibit the single SUMO-activating enzyme (SAE1–SAE2) and thus all cellular SUMOylation processes[11,12], now paves the way for comprehensive exploration of the consequences of blocking SUMO signaling and the underlying mechanisms. This may instruct therapeutic opportunities for pharmacological intervention of the SUMO modification pathway, a promising new avenue in cancer treatment[13]. In the present study, motivated by this potential, we performed genome-scale clustered regularly interspaced short palindromic repeats (CRISPR)–Cas9 screens to systematically profile the genetic vulnerabilities to inhibition of SUMOylation in human cells. These and subsequent CRISPR screens revealed synthetic lethality (SL) relationships between SUMO signaling, the BTRR-PICH (Plk1-interacting checkpoint helicase) pathway that resolves UFBs arising from catenated DNA structures persisting into mitosis, and the poorly characterized protein NFAT-interacting protein 45 (NIP45). Based on these genetic interactions, we discovered that SUMO and NIP45 orchestrate an interphase pathway for resolving DNA catenanes by promoting their conversion into double-strand breaks (DSBs), which together with the mitotic BTRR-PICH pathway for UFB resolution provides an indispensable cellular barrier toward mitotic failure caused by toxic catenated DNA structures generated during a normal cell-cycle.

## Results

### Genetic vulnerabilities to inhibition of SUMOylation

To comprehensively identify genetic vulnerabilities to inhibition of SUMOylation in human cells, we performed genome-scale CRISPR–Cas9 dropout screens for genes whose targeted knockout (KO) confers hypersensitivity to SUMOi. To this end, cells infected with the TKOv3 single guide (sg)RNA library targeting 18,053 protein-coding human genes[14] were grown for several population doublings in the absence or presence of a low dose of SUMOi corresponding to 20% of the lethal dose ($LD_{20}$), which moderately reduces, but does not abolish, overall SUMOylation activity (Fig. 1a and Extended Data Fig. 1a–c). Concurrent screens using HeLa cervical cancer cells and nontransformed retinal pigment epithelial 1 (RPE1) cells revealed 74 and 53 genes, respectively, whose KO hypersensitizes cells to SUMOi treatment (Fig. 1b–d and Supplementary Data 1 and 2). Surprisingly, considering the widespread involvement of SUMO in cellular signaling processes, only eight screen hits were shared between the two cell lines (Fig. 1b–d and Extended Data Fig. 1d). This suggests considerable cell type-specific differences in the relative importance of SUMO-mediated processes in supporting fitness and that only a small core set of proteins is instrumental in sustaining cell proliferation when SUMOylation is impaired during otherwise unperturbed growth.

Besides *ABCG2*, which encodes a multidrug transporter that probably extrudes SUMOi from cells and *SAE1* encoding the noncatalytic subunit of the SUMO E1 enzyme heterodimer, the shared hits were *RMI1*, *RMI2*, *NFATC2IP*, *EP300*, *CRAMP1L* and *FKBP8* (Fig. 1b,c and Extended Data Fig. 1d). SUMOi hypersensitivity resulting from individual small interfering (si)RNA-mediated depletion of these factors was validated in proliferation assays (Fig. 1e,f and Extended Data Fig. 1e). However, overall SUMOylation levels were not reduced by these knockdowns, unlike the expected impact of SAE1 depletion (Extended Data Fig. 1b,c). RMI1 and RMI2 are both integral components of the BTRR complex, which has a central role in disentangling catenated DNA structures arising from DNA replication and repair intermediates during interphase and cooperates with the DNA helicase PICH to resolve UFBs that form when interlinked DNA structures persist into mitosis[15]. In support of a functional relevance of the latter involvement in protecting against SUMOi, KO of both *BLM* and *ERCC6L* (encoding PICH) conferred hypersensitivity to SUMOi in HeLa cells, as did BLM (Bloom syndrome protein) and PICH knockdown (Fig. 1b,c,e). Moreover, although *BLM* was slightly below the significance threshold in RPE1 cells (NormZ value −2.58; Supplementary Data 2) and *ERCC6L* is essential in this background[16], RPE1 BLM–KO cells[17] displayed strong hypersensitivity

to SUMOi relative to their wild-type (WT) counterparts (Extended Data Fig. 1f). Importantly, loss of RMI1 or RMI2 was epistatic with PICH depletion in sensitizing cells to SUMOi (Fig. 1g and Extended Data Fig. 1g–j). This suggests that the requirement for the BTRR complex in protecting against SUMOi cytotoxicity mainly entails its mitotic function in resolving UFBs together with PICH. Collectively, these findings identify a core set of proteins including BTRR and PICH that become essential for cell proliferation when SUMOylation is compromised.

### Synthetic lethality relationships between SUMOi, NIP45 and BTRR-PICH

We next focused on *NFATC2IP* (encoding the protein NIP45), the KO of which conferred the strongest hypersensitivity to SUMOi in both the RPE1 and the HeLa screens (Fig. 1b,c). Generation of NIP45–KO cell lines confirmed exquisite SUMOi hypersensitivity resulting from NIP45 deficiency in RPE1, HeLa and U2OS osteosarcoma cells (Fig. 1h–j, Extended Data Fig. 2a and Fig. 2c,d). Loss of NIP45 also strongly hypersensitized cells to the SUMOylation inhibitor TAK-981, a functional analog of the ML-792 SUMOi displaying promising clinical potential[12] (Extended Data Fig. 2b). Notably, loss of NIP45 had no significant impact on cell proliferation and cell-cycle status in unperturbed cells but led to strongly impaired proliferation upon low-dose SUMOi treatment (Fig. 1k and Extended Data Fig. 2c). Depletion of EP300, CRAMP1 or FKBP8 was nonepistatic with NIP45–KO in sensitizing cells to SUMOi (Extended Data Fig. 2d), suggesting that these factors function independently of NIP45 in promoting cell fitness when SUMOylation is compromised. NIP45 is highly conserved through eukaryotic evolution and all known orthologs contain tandem carboxy-terminal SUMO-like domains (SLDs) that are unique to this family of proteins (Fig. 2a)[18,19]. Whereas both the *Saccharomyces cerevisiae* and *Schizosaccharomyces pombe* NIP45 orthologs (Esc2 and Rad60, respectively) have been implicated in genome stability maintenance via their SLDs[20–26], no corresponding role for human NIP45 has been reported. Indeed, loss of NIP45 had no discernible impact on the sensitivity to a range of genotoxic agents (Extended Data Fig. 2e). Stable reconstitution of NIP45–KO cells with green fluorescent protein (GFP)-tagged WT NIP45 fully rescued proliferation in the presence of SUMOi (Fig. 2b–d). By contrast, complementation with GFP–NIP45 mutants lacking either of the SLD domains (ΔSLD1 and ΔSLD2) or containing a point mutation (Asp394Arg; SLD2*) predicted to functionally inactivate the SLD2 domain, based on its crystal structure and homology to a previously described corresponding Rad60 mutant[21], failed to appreciably restore proliferation upon SUMOi treatment, despite both the WT and the mutant GFP–NIP45 proteins localized diffusely to the nucleus like endogenous NIP45 (Fig. 2b–d and Extended Data Fig. 2f,g). Thus, both the SLD1 and the SLD2 domains are essential for the role of NIP45 in underpinning proliferation when SUMOylation is impaired. Analysis of NIP45–KO cells reconstituted with GFP–NIP45 proteins lacking part or all of the sequence amino terminal to the SLDs (Fig. 2b and Extended Data Fig. 2h) showed that a predicted α-helix in proximity to SLD1 is also critical for the ability of NIP45 to preserve growth in the presence of SUMOi (Extended Data Fig. 2i).

To understand the cellular function of NIP45 and the basis for its selective essentiality upon impairment of SUMOylation, we performed parallel genome-scale CRISPR–Cas9 screens for SL relationships with NIP45–KO in RPE1 and HeLa backgrounds (Fig. 2e, Fig. 1h and Extended Data Fig. 2j). Notably, among gene KOs that selectively impaired proliferation of NIP45–KO cells, *RMI1*, *RMI2* and *BLM* were the only hits shared between the HeLa and RPE1 screens along with *SYS1*, which encodes a Golgi trafficking protein whose possible significance in promoting proliferation in the absence of NIP45 is unclear (Fig. 2f,g and Supplementary Data 3 and 4). In addition, NIP45 deficiency was synthetic lethal with PICH KO in HeLa cells (Fig. 2f). This suggested that, in addition to being individually required for survival in SUMOi-treated cells, combined loss of NIP45 and BTRR-PICH function is incompatible

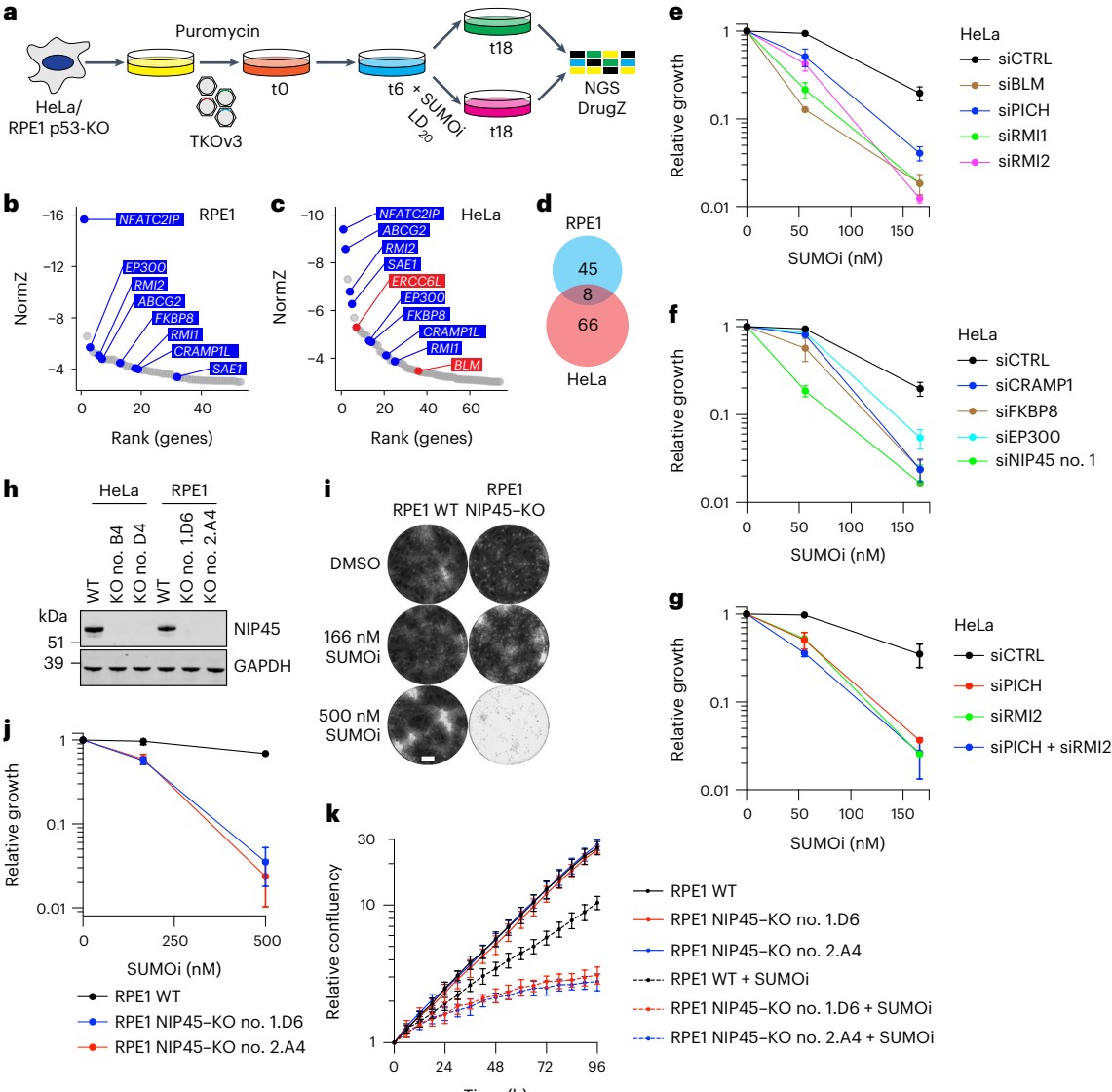

**Fig. 1 | System-wide mapping of genetic vulnerabilities to inhibition of SUMOylation in human cells. a**, Schematic outline of genome-scale CRISPR–Cas9 screens for genes whose KO sensitizes cells to SUMOi. NGS, next-generation sequencing. **b,c**, DrugZ analysis of sgRNA depletion in RPE1 cells (**b**) and HeLa cells (**c**) after 12 d of low-dose SUMOi treatment (56 nM) ($n$ = 2 technical replicates). A NormZ value of <−3 was used as the cut-off for defining significant genes. Hits common to both screens are highlighted in blue; hits that form a complex with common genes are highlighted in red. *NFATC2IP* encodes NIP45 and *ERCC6L* encodes PICH (see Supplementary Data 1 and 2 for full results). **d**, Venn diagram of significant genes (NormZ <−3) from DrugZ analysis of CRISPR screens in (**b** and **c**). **e**, SRB cell growth assay using HeLa cells treated with non-targeting control (CTRL), BLM, PICH, RMI1 or RMI2 siRNAs and indicated SUMOi doses (mean ± s.d.; $n$ = 3 independent experiments). **f**, As in (**e**), using CTRL, CRAMP1,

FKBP8, EP300 or NIP45 siRNAs (mean ± s.d.; $n$ = 3 independent experiments). **g**, As in (**e**), using CTRL, PICH and/or RMI2 siRNAs (mean ± s.d.; $n$ = 3 independent experiments). **h**, Western blot analysis of NIP45 protein levels in whole-cell lysates from indicated cell lines. Data represent two independent experiments with similar outcome. GAPDH, Glyceraldehyde 3-phosphate dehydrogenase. **i,j**, Representative images (**i**) and quantification (**j**) of SRB cell growth assay using RPE1 WT and NIP45–KO cells treated with indicated SUMOi doses (mean ± s.d.; $n$ = 3 independent experiments). DMSO, Dimethylsulfoxide. Scale bar, 0.25 cm (**i**). **k**, Incucyte cell growth assay measuring cell density of RPE1 WT and NIP45–KO cells treated continuously with indicated SUMOi doses (mean ± s.d.; $n$ = 4 independent experiments). Images were acquired every 6 h for 4 d and data points fitted to nonlinear exponential growth models.

with cell proliferation. Validating this notion, we observed dramatic loss of viability when BLM or PICH was knocked down in NIP45–KO cells in the absence of SUMOi treatment (Fig. 2h,i). The proliferation defect of PICH-depleted NIP45–KO cells could be rescued by stably expressed WT NIP45 but not the SLD2* mutant (Extended Data Figure 2k). Knockdown of NIP45 in otherwise untreated RMI1–KO cells also led to a strong block to proliferation (Extended Data Fig. 2l). Together with the above findings, these data reveal strong SL relationships between SUMO signaling, BTRR-PICH function and NIP45, involving its SLD domains (Fig. 2j).

## NIP45 and SUMO guard against UFB formation and binucleation

Given the established key role of BTRR-PICH in resolving UFBs in mitosis[15], we surmised that NIP45 loss and inhibition of SUMOylation might be detrimental to cells lacking BTRR-PICH function by impacting upon UFB formation and/or resolution. Consistent with this idea, both the average number of UFBs per cell and the proportion of UFB-positive cells were significantly elevated in U2OS NIP45–KO cells (Fig. 3a,b). A comparable effect was seen upon low-dose SUMOi treatment of parental cells (Fig. 3a,b). Notably, combined NIP45 loss

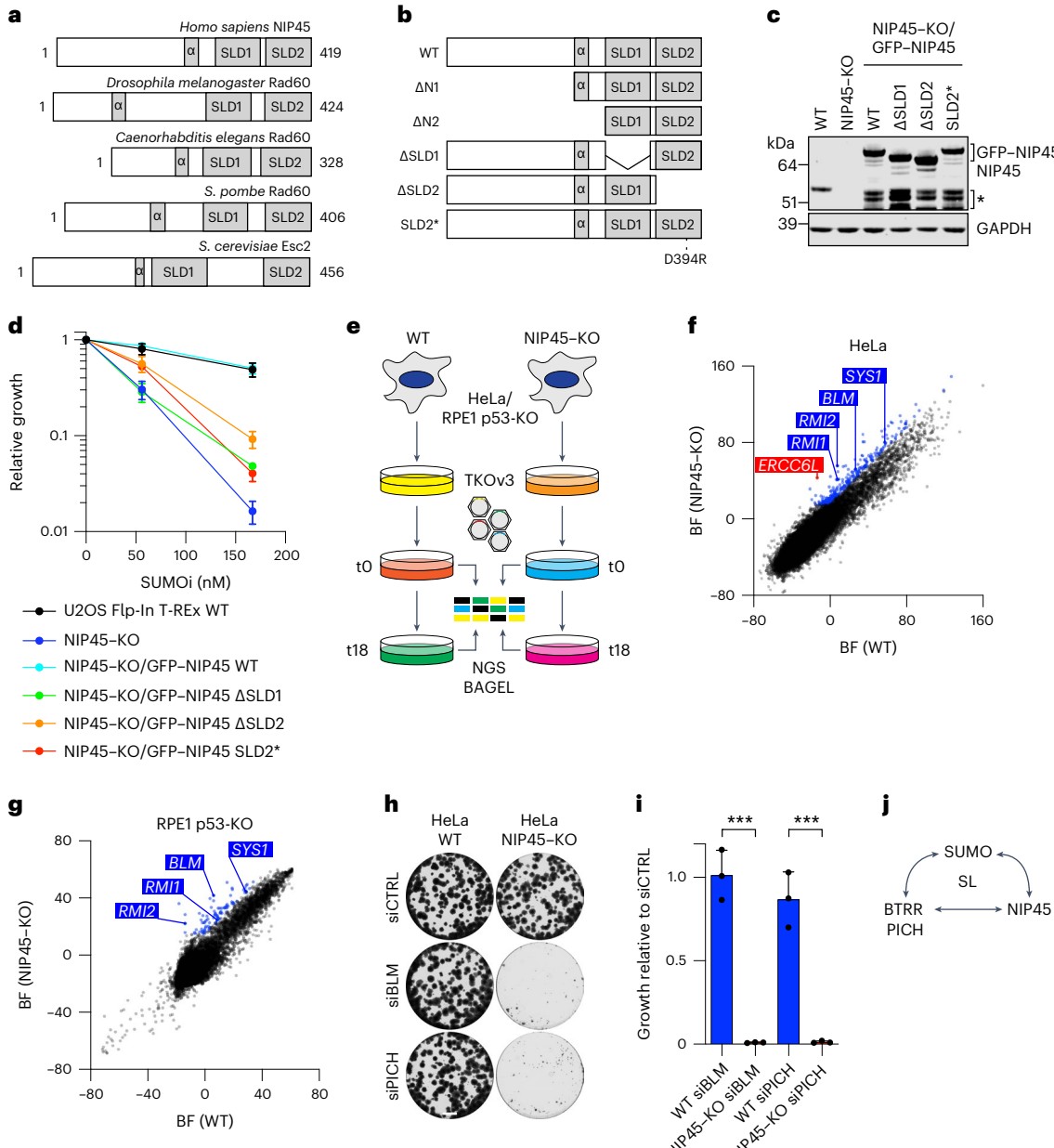

**Fig. 2 | NIP45 and its SLDs are essential for proliferation in the absence of SUMOylation or BTRR-PICH function. a**, Domain organization of indicated eukaryotic NIP45 orthologs showing conservation of the tandem SLDs and a predicted N-terminal α-helix. **b**, Schematic showing WT and mutant human NIP45 proteins analyzed in the present study. **c**, Western blot analysis of NIP45 protein levels in whole-cell lysates from U2OS Flp-In T-Rex WT and NIP45-KO cell lines expressing indicated exogenous GFP–NIP45 variants (**b**). Bands marked by an asterisk represent breakdown products of GFP–NIP45. **d**, SRB cell growth assay using U2OS Flp-In T-Rex WT and NIP45-KO cell lines stably expressing indicated GFP–NIP45 variants that were treated with indicated SUMOi doses (mean ± s.d.; *n* = 3 independent experiments). **e**, Schematic outline of genome-scale CRISPR–Cas9 screens for SL in NIP45-KO cell lines. **f,g**, BAGEL analysis of

sgRNA depletion in HeLa cells (**f**) and RPE1 cells (**g**) comparing WT and NIP45-KO cell lines (*n* = 2 technical replicates). Synthetic lethal genes common to both screens are highlighted in blue and genes in complex with hits common to both screens but scoring in one screen only are highlighted in red (see Supplementary Data 3 and 4 for the full results). **h**, SRB cell growth assay using HeLa WT and NIP45-KO cells treated with the indicated siRNAs. Scale bar, 0.25 cm. **i**, SRB cell growth assay using HeLa WT and NIP45-KO cells treated with indicated siRNAs (mean ± s.d.; *n* = 3 independent experiments; unpaired two-tailed Student's *t*-test; siBLM: ***P* = 0.0003; siPICH: ***P* = 0.0009). **j**, Schematic representation of SL relationships between NIP45, SUMO signaling and BTRR-PICH. Data represent three (**h**) and two (**c**) independent experiments with similar outcome.

and low-dose SUMOi exposure exacerbated UFB accumulation, leading to virtually all cells manifesting with multiple UFBs even though other mitotic chromosome abnormalities were absent (Fig. 3a,b). Similar effects were observed in HeLa cells (Extended Data Fig. 3a). Complementation of NIP45-KO cells with GFP–NIP45 WT restored UFB formation to levels seen in parental cells (Fig. 3b). UFBs are known

to accumulate after replication stress induced by treatment with the replicative DNA polymerase inhibitor aphidicolin and upon catalytic inhibition of topoisomerase II (TOP2), which resolves double-stranded DNA (dsDNA) catenanes[15]. Importantly, however, we found that NIP45 deficiency led to increased UFB levels after treatment with the TOP2 catalytic inhibitor ICRF-193, but not aphidicolin (Fig. 3c), suggesting

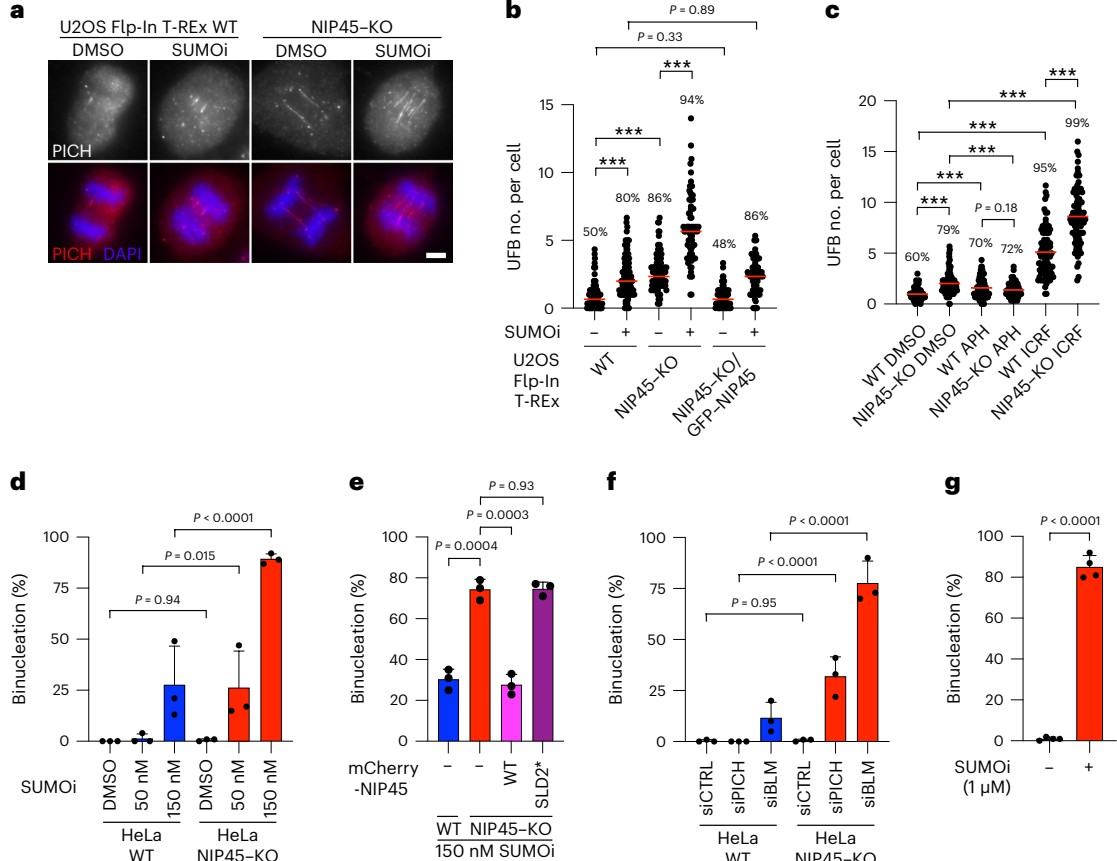

**Fig. 3 | NIP45 and SUMOylation protect against excessive UFB formation and binucleation. a**, Representative immunofluorescence images of U2OS Flp-In T-REx WT and NIP45–KO cells immunostained with PICH antibody (red) to identify UFBs after treatment with SUMOi (50 nM) for 24 h. Scale bar, 5 μm. **b**,**c**, Quantification of UFBs in in U2OS Flp-In T-REx WT, NIP45–KO and NIP45–KO/ GFP–NIP45 cells after treatment with SUMOi (50 nM) (**b**), aphidicolin (APH; 0.4 μM) and ICRF-193 (ICRF; 0.1 μM) (**c**) for 24 h (all data points are shown; red bars, median; *n* = 3 independent experiments; at least 80 cells scored per condition per independent experiment; unpaired, two-tailed Student's *t*-test;

***P < 0.0001). The percentage above the bars indicates the fraction of cells containing at least one UFB (UFB-positive cells). **d**–**g**. Quantification of live-cell imaging tracking the mitotic fate of HeLa WT, NIP45–KO and NIP45–KO cells transiently expressing mCherry–NIP45 WT or SLD2* after 48 h of pre-treatment with indicated doses of SUMOi (**d**, **e** and **g**) or transfected with indicated siRNAs (**f**) (mean ± s.d.; *n* = 3 independent experiments; at least 40 cells (**d**), 25 cells (**e**), 35 cells (**f**) and 50 cells (**g**) were scored per condition per replicate; one-way analysis of variance (ANOVA) without adjustment for multiple comparisons for **d** and **f** and unpaired, two-tailed Student's *t*-test for **e** and **g**).

that NIP45 may be particularly important for preventing accumulation of UFBs arising from unresolved dsDNA catenanes rather than from under-replicated DNA.

Failure to resolve UFBs can interfere with faithful chromosome segregation, which in some cases leads to abortive cytokinesis and binucleation[27]. Live-cell imaging analysis showed that SUMOi treatment caused a dose-dependent increase in binucleation frequency after mitosis in parental HeLa cells but had no impact on the kinetics of mitotic progression (Fig. 3d and Extended Data Fig. 3b). Remarkably, although NIP45–KO on its own did not significantly impact mitotic progression and binucleation, the rate of SUMOi-induced binucleation was greatly enhanced by NIP45 deficiency; in fact, approximately 80% of all cell-division attempts resulted in cytokinesis failure leading to binucleation when NIP45–KO cells were treated with a moderate (150 nM) SUMOi dose (Fig. 3d and Extended Data Fig. 3c,d), consistent with the strong increase in UFB formation in SUMOi-treated NIP45–KO cells (Fig. 3a,b and Extended Data Fig. 3a). These cells typically initiated cytokinesis but remained connected by a visible intercellular bridge for an extended amount of time before coalescing into a binucleated cell (Supplementary Video 1). The impact of NIP45 KO on binucleation could be rescued by complementation with WT NIP45 but not the SLD2* mutant (Fig. 3e). Moreover, whereas knockdown of BLM or PICH alone had limited impact on binucleation frequency, a substantial proportion

of dividing cells became binucleated on depletion of either factor in a NIP45-deficient background (Fig. 3f). Complete inhibition of SUMOylation by high-dose SUMOi treatment led to binucleation after mitotic exit in approximately 80% of parental cells, mirroring the impact of functional BTRR-PICH inactivation or partial SUMOylation impairment in NIP45–KO cells (Fig. 3d,f,g). This raises the possibility that the role of SUMOylation in suppressing UFB formation and ensuing binucleation is exerted via nonepistatic NIP45- and BTRR-PICH-driven mechanisms. In support of this proposal, SUMOi, but not NIP45 loss, hypersensitized cells to ICRF-193 (Extended Data Figs. 2e and 3e). Collectively, these data show that NIP45 and SUMOylation are required for preventing excessive UFB formation, which leads to binucleation accompanied by diminished proliferative potential when BTRR-PICH-dependent UFB resolution is defective, providing a rationale for the SL relationships between NIP45, SUMOylation and BTRR-PICH (Fig. 2j).

## NIP45 and SUMO induce G2 arrest upon decatenation inhibition

We next addressed how NIP45 and SUMOylation counteract UFB accumulation and subsequent binucleation. Unlike BTRR-PICH components, we observed no detectable NIP45 association with UFBs or chromatin (Extended Data Figs. 2f and 3f), arguing against a direct role of NIP45 in UFB resolution. Moreover, the lack of hypersensitivity of NIP45–KO cells

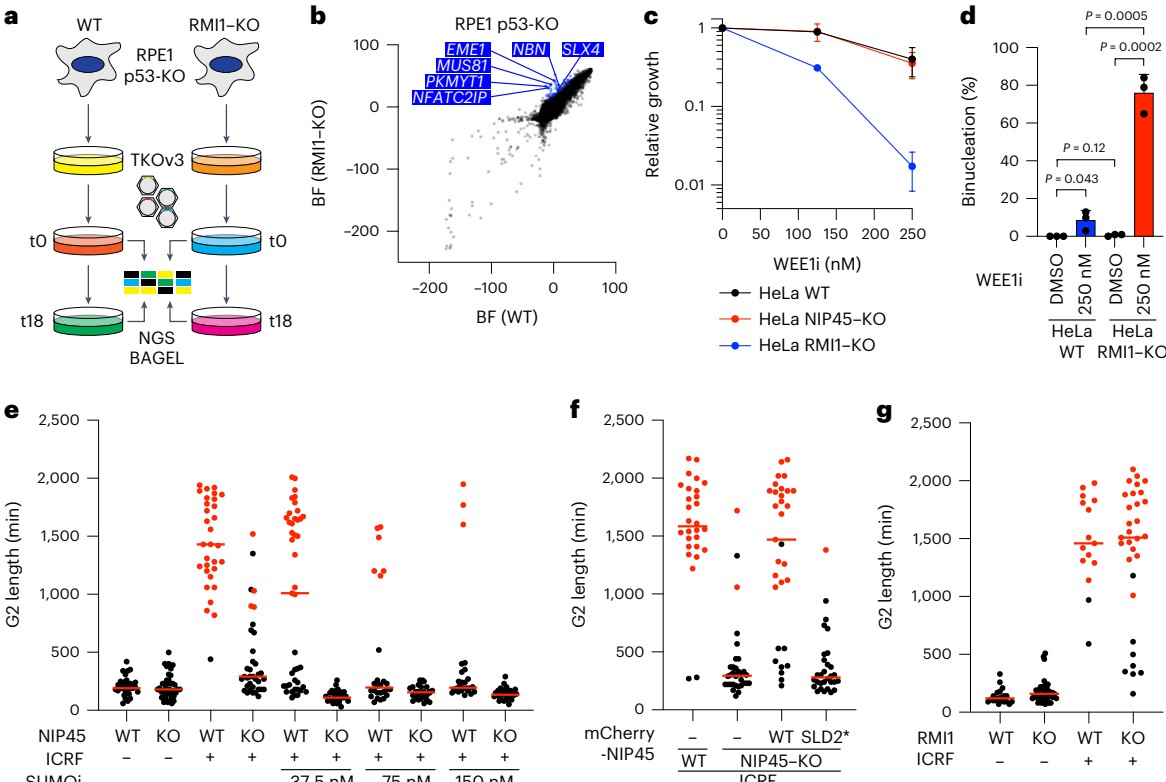

**Fig. 4 | NIP45 and SUMOylation are required for G2 arrest upon inhibition of TOP2-dependent decatenation. a**, Schematic outline of genome-scale CRISPR–Cas9 screen for SL relationships in RPE1 RMI1–KO/p53-KO cells. **b**, BAGEL analysis of sgRNA depletion comparing WT and RMI1–KO cell lines (*n* = 2 technical replicates). Synthetic lethal genes (blue) are indicated (see Supplementary Data 5 for full results). **c**, SRB cell growth assay using HeLa WT, NIP45–KO and RMI1–KO cells treated with indicated WEE1i doses (mean ± s.d.; *n* = 3 independent experiments). **d**, Quantification of live-cell imaging tracking the mitotic fate of HeLa WT and RMI1–KO cells after 24 h of pre-treatment with WEE1i (250 nM) (mean ± s.d.; *n* = 3 independent experiments; at least 50 cells were scored per condition per replicate; unpaired, two-tailed Student's *t*-test). **e,f**, Quantification of live-cell imaging to analyze G2 length (defined as the time from disappearance

of GFP–PCNA foci to nuclear envelope breakdown) in HeLa WT and NIP45–KO cells (**e**) complemented with mCherry–NIP45 WT and SLD2* (**f**) after pre-treatment for 48 h with indicated SUMOi doses and exposed to ICRF-193 (7 μM) immediately before imaging (red bars, median; representative experiment of *n* = 3 independent experiments; at least 30 cells (**e**) and 28 cells (**f**) were scored per condition per replicate). Red dots denote cells that did not enter mitosis during the experiment. **g**, Quantification of live-cell imaging to analyze G2 length in HeLa WT and RMI1–KO cells exposed to ICRF-193 (ICRF; 7 μM) immediately before filming (red bars, median; representative experiment of *n* = 3 independent experiments; at least 15 cells were scored per condition per replicate). Red dots denote cells that did not enter mitosis during the experiment.

to DNA damage- and replication stress-inducing agents suggested that NIP45 deficiency does not lead to elevated UFB formation by increasing the level of unresolved replication or recombination intermediates (Extended Data Fig. 2e). We reasoned that identifying genes required for survival of cells lacking BTRR-PICH function might provide clues to how NIP45 prevents excessive UFB formation. We therefore carried out a CRISPR–Cas9 screen for genes whose ablation is lethal in RPE1 RMI1–KO cells, which are deficient for BTRR-PICH-mediated UFB resolution (Fig. 4a and Extended Data Fig. 4a). Consistent with and corroborating our results above, *NIP45* was among the strongest hits in this screen (Fig. 4b and Supplementary Data 5). Further validating the screen, we observed SL between RMI1 KO and loss of the multi-nuclease scaffold protein SLX4 or the associated MUS81–EME1 nuclease complex (Fig. 4b), as has been reported previously[28]. Interestingly, the screen also revealed that KO of *PKMYT1*, which encodes the MYT1 kinase that restricts mitotic entry via inhibitory phosphorylation of CDK1[29,30], is synthetic lethal with RMI1 deficiency (Fig. 4b). The critical importance of intact G2/M control for underpinning cell proliferation in the absence of RMI1 was validated using a well-established, small-molecule inhibitor of the WEE1 kinase (WEE1i)[31], which catalyzes inhibitory phosphorylation of CDK1 together with MYT1 (refs. 32,33) (Fig. 4c). Notably, treatment of otherwise unperturbed RMI1–KO cells, but not parental cells, with WEE1i led to extensive binucleation, phenocopying the effect of

combined loss of NIP45 and BTRR-PICH function (Figs. 4d and 3f). By contrast, NIP45 KO did not sensitize cells to WEE1i treatment (Fig. 4c). These findings show that preventing premature entry into mitosis is crucial for faithful cell division and proliferation in the absence of BTRR.

The observations above raised the possibility that NIP45 and SUMOylation may counteract UFB accumulation and binucleation by restraining mitotic entry in the presence of unresolved DNA entanglements. To test this idea, we treated cells with ICRF-193 to induce dsDNA catenane accumulation and analyzed the impact on G2/M transition kinetics using live-cell imaging. In parental cells, ICRF-193 exposure led to an extensive delay in the timing of mitotic entry (Fig. 4e and Extended Data Fig. 4b,c), in agreement with a previously reported response that restricts G2/M transition when TOP2-dependent dsDNA catenane resolution is blocked by ICRF-193 (referred to by some studies as the 'decatenation checkpoint'), but whose precise molecular basis is unclear[34–37]. Strikingly, however, this G2 delay was strongly diminished in NIP45–KO cells (Fig. 4e and Extended Data Fig. 4c). Inhibiting SUMOylation also drastically attenuated ICRF-193-induced G2 arrest in a dose-dependent manner and low-dose SUMOi treatment was sufficient to eliminate this response in NIP45–KO cells, paralleling the impact on binucleation (Figs. 3d and 4e and Extended Data Fig. 4c,d). Consequently, virtually all cells that bypassed ICRF-193-induced G2 arrest due to NIP45 deficiency or SUMOi treatment became binucleated, as expected

from the requirement of TOP2 activity for chromosome segregation (Extended Data Fig. 4d,e). The defective ICRF-193-induced G2/M arrest in NIP45–KO cells was not due to altered TOP2A expression and could be restored by complementation with WT NIP45 but not SLD-mutated alleles (Fig. 4f and Extended Data Fig. 4f,g). RMI1–KO cells displayed intact G2/M arrest upon ICRF-193 treatment (Fig. 4g), suggesting that although BTRR is instrumental for processing UFBs it is not required for delaying mitotic entry in the presence of unresolved catenanes. Unlike ICRF-193-induced G2 arrest, NIP45 was dispensable for the canonical G2 DNA-damage checkpoint triggered by ionizing radiation (IR)-induced DSB formation (Extended Data Fig. 4h). Collectively, these data show that NIP45 and SUMOylation are required for a G2 cell-cycle checkpoint response restraining mitotic entry when TOP2-dependent catenane resolution is blocked, explaining why loss of NIP45 or impairment of SUMO signaling leads to UFB accumulation.

## SUMO and NIP45 promote DNA catenane conversion into DSBs

Prompted by the above findings, we asked how NIP45 and SUMOylation promote G2 arrest when TOP2-dependent decatenation is inhibited by ICRF-193. Although catalytic inhibition of TOP2 should, in principle, not give rise to DNA breakage, unlike TOP2 poisons such as etoposide, some previous studies (for example, ref. [38]) provided evidence for DSB formation upon ICRF-193 treatment. In line with this, we noted that the SUMO- and NIP45-dependent cell-cycle arrest triggered by ICRF-193 treatment was accompanied by a moderate, but consistent, induction of multiple markers of DSB formation, including γH2AX and 53BP1 foci demarcating DSB sites, autophosphorylation denoting activation of the ATM (ataxia–telangiectasia mutated) kinase, a master organizer of the DSB response, as well as upregulation of ATM-dependent phosphorylation sites in CHK2 and KAP1 (Fig. 5a,b and Extended Data Fig. 5a,b). Moreover, our RMI1–KO CRISPR screen revealed that, similar to NIP45, NBS1, a component of the MRE11-RAD50-NBS1 (MRN) complex that promotes ATM activation on DSB formation[39], was synthetic lethal with loss of BTRR function (Fig. 4b), and NBS1 knockdown abolished CHK2 phosphorylation upon ICRF-193 treatment (Extended Data Fig. 5c). In agreement with this, ICRF-193-induced formation of γH2AX foci and G2 arrest was impaired by inhibition of ATM and, to a lesser extent, ATR (ATM and Rad3-related), as reported previously[36] (Extended Data Fig. 5d,e). Importantly, neutral comet assays provided direct evidence for DSB formation upon ICRF-193 treatment (Fig. 5c,d and Extended Data Fig. 5f,g). Blocking mitotic exit by nocodazole addition did not affect ICRF-193-induced DSB signaling (Extended Data Fig. 5h), ruling out that these lesions are generated by physical breakage of UFBs during cytokinesis, as has been reported for UFBs arising from homologous recombination intermediates[40]. These observations suggested that the G2 arrest in response to ICRF-193-induced dsDNA catenane accumulation is a consequence of DSB formation, and we surmised that NIP45 and SUMOylation might be required for generating these

breaks. Indeed, consistent with the impact on G2/M transition kinetics, NIP45 deficiency greatly reduced the accumulation of DSBs and associated markers upon ICRF-193 treatment, and this could be rescued by complementation of NIP45–KO cells with WT but not SLD-mutated forms of GFP–NIP45 (Fig. 5a–c and Extended Data Fig. 5a,b,i,j). Inhibition of SUMOylation by ML-792 or TAK-981 quantitatively suppressed ICRF-193-dependent DSB generation and signaling, paralleling our observations on ICRF-193-induced G2 arrest (Figs. 4e and 5a,b,d and Extended Data Fig. 5g,k,l). Importantly, however, the requirement of NIP45 and SUMOylation for DSB induction and signaling was specific to inhibition of TOP2-dependent decatenation, since NIP45 KO or SUMOi treatment had no impact on DSB signaling elicited by a panel of DNA damage- and replication stress-inducing agents, including the TOP2 poison etoposide and IR that generate DSBs directly (Fig. 5a and Extended Data Fig. 5m). Taken together, these findings suggest that NIP45 and SUMOylation orchestrate an interphase pathway for converting dsDNA catenanes into DSBs, thereby triggering G2 arrest via canonical ATM/ATR-dependent DNA-damage signaling.

## NIP45-dependent SUMOylation of catenane cleavage components

Yeast NIP45 orthologs interact with the posterior face of the SUMO E2 enzyme UBC9 via the SLD2 domain and, by means of this association, have been suggested to function as cofactors for specific SUMOylation processes[20,41,42]. We found that human NIP45 also binds UBC9 and proteomic analysis showed that the SLD2* point mutation specifically abrogates this interaction (Fig. 5e, Extended Data Fig. 6a and Supplementary Data 6). We confirmed biochemically that both the NIP45 ΔSLD2 and SLD2* mutants were deficient for binding to UBC9, whereas deletion of SLD1 had no impact (Extended Data Fig. 6b). As the ΔSLD2 and SLD2* mutants both failed to rescue any NIP45-mediated phenotype that we observed, this strongly suggests that NIP45 promotes catenane processing and ensuing DSB formation on ICRF-193 treatment by stimulating the SUMOylation of one or more effector proteins via its interaction with UBC9. To identify such factors, we used mass spectrometry (MS) to profile global SUMOylation changes resulting from NIP45 loss or ICRF-193 exposure (Fig. 5f and Extended Data Fig. 6c). Consistent with our observation that NIP45 loss had no detectable impact on total cellular SUMOylation levels (Extended Data Fig. 1b), this analysis showed that the great majority of SUMO target proteins displayed unaltered SUMOylation status upon NIP45 depletion (Fig. 5f and Supplementary Data 7). However, we identified a small subset of proteins showing strongly reduced SUMOylation in cells lacking NIP45 (Fig. 5f and Supplementary Data 7). Interestingly, the NIP45-dependent SUMOylation targets comprised both SLX4 and EME1, a binding partner of the SLX4-associated nuclease MUS81 (Fig. 5f). By contrast, although TOP2 SUMOylation has been suggested to be functionally relevant for the decatenation checkpoint[36,43], our MS

**Fig. 5 | NIP45 promotes DNA catenane conversion into DSBs involving SUMOylation of the SLX4 multi-nuclease complex. a,** Western blot analysis of whole-cell lysates from HeLa WT and NIP45–KO cells treated for 2 h with ICRF-193 (7 μM), SUMOi (2 μM) and/or IR (4 Gy). **b,** Immunofluorescence analysis of γH2AX foci in U2OS Flp-In T-REx WT and NIP45–KO cells after treatment with ICRF-193 (1 μM) and/or SUMOi (2 μM) for 4 h (mean ± s.d.; $n = 3$ independent experiments; unpaired, two-tailed Student's $t$-test). **c,d,** DSBs (tail moment) analyzed by neutral comet assay in HeLa WT and NIP45–KO cells treated with ICRF-193 (ICRF; 25 μM) (**c**) and/or SUMOi (2 μM) (**d**) for 2 h (black bars, median; $n = 3$ independent experiments; at least 50 cells were scored per condition per replicate; unpaired, two-tailed Mann–Whitney $U$-test; ***$P < 0.0001$). **e,f,** MS analysis of GFP pulldowns from U2OS Flp-In T-REx NIP45–KO cells stably expressing GFP–NIP45 wt or SLD2* (**e**) or SUMOylated proteins isolated by denaturing His (Ni-NTA) pulldown from HeLa or HeLa/His₁₀-SUMO2 cells transfected with indicated siRNAs (**f**). Volcano plots show the mean difference of the protein intensity plotted against the $P$ value (two-tailed, two-sample Student's $t$-test). Significant differences

($q < 0.05$) were calculated by permutation-based false discovery rate (FDR) control (2,500 rounds of randomization) and are indicated in blue ($n = 4$ biological replicates). See Supplementary Data 6 and 7 for full results. LC/MS-MS, Liquid chromatography–tandem MS. **g,** Western blot analysis of denaturing His (Ni-NTA) pulldown from HeLa or HeLa/His₁₀-SUMO2 cells transfected with the indicated siRNAs. **h,** Western blot analysis of GFP immunoprecipitates from whole-cell lysates of HEK293 cells transfected with plasmids encoding GFP–NIP45 WT and Flag-HA-SLX4. **i,** Western blot analysis of whole-cell lysates from HeLa cells transfected with indicated siRNAs and treated with ICRF-193 (7 μM) for 2 h. **j,** Immunofluorescence analysis of γH2AX foci in HeLa cells treated with control or SLX4 siRNAs and subjected to treatment with ICRF-193 (1 μM) for 4 h (mean ± s.d.; $n = 3$ independent experiments; unpaired, two-tailed Student's $t$-test). **k,** Model of SUMO-mediated resolution of toxic DNA catenanes via nonepistatic NIP45- and BTRR-PICH-dependent pathways (see main text for details). Data represent three (**a** and **h**) and two (**g** and **i**) independent experiments with a similar outcome.

experiments showed that neither NIP45 knockdown nor ICRF-193 exposure significantly affected TOP2 SUMOylation levels (Fig. 5f, Extended Data Fig. 6c and Supplementary Data 7), indicating that TOP2 is not a primary target of NIP45-dependent G2 arrest via SUMOylation. We confirmed biochemically that SUMOylation of both SLX4 and EME1 is strongly reduced in NIP45-deficient cells (Fig. 5g and Extended Data Fig. 6d) and that EME1 SUMOylation depends on SLX4, in line with previous observations[44] (Extended Data Fig. 6e,f). Thus, NIP45 is critically required for SUMOylation of SLX4 and associated nuclease components. Nevertheless, although NIP45 promotes ICRF-193-induced G2

arrest in a manner requiring UBC9 binding via SLD2, ICRF-193 exposure did not significantly alter the SUMOylation of SLX4, EME1 and other proteins, whose SUMO modification is stimulated by NIP45 (Extended Data Fig. 6c). This suggests that NIP45-dependent SUMOylation processes are fully operational during an unperturbed cell cycle, consistent with the SL relationships between NIP45, SUMO and BTRR-PICH in the absence of exogenous insults. Given that NIP45 binds UBC9 via SLD2 and is required for SUMOylation of specific factors, we considered the possibility that it might function as a SUMO E3 ligase. We observed that, similar to the SUMO E3 ligase RanBP2, recombinant NIP45 underwent

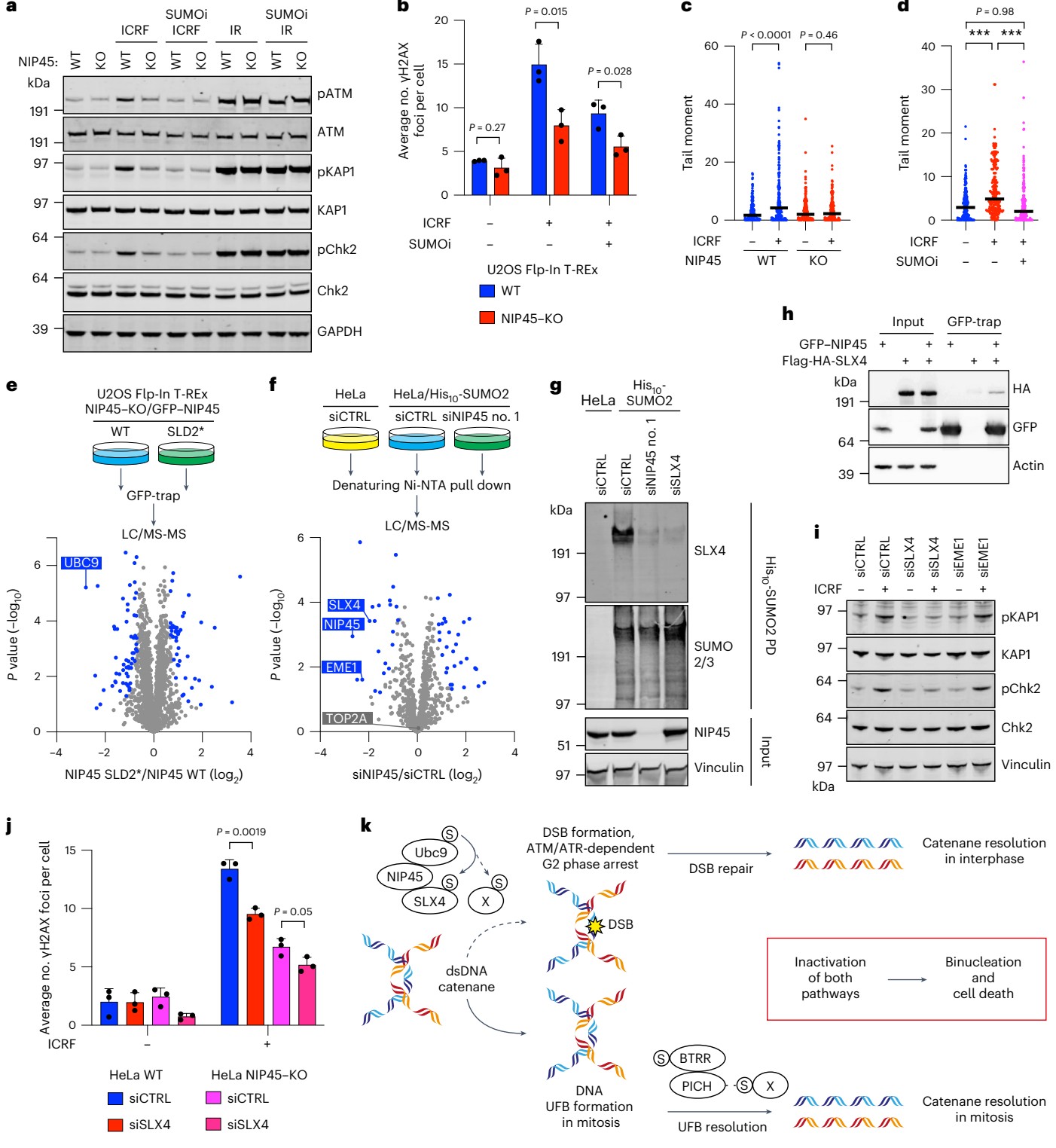

extensive auto-SUMOylation in the presence of UBC9 in vitro, in a reaction that was considerably more efficient than the modification of the optimal SUMO substrate SP100 under the same assay conditions (Extended Data Fig. 7a–c). Interestingly, although NIP45 promoted polymerization of both free SUMO1 and SUMO2, it was much more efficient at modifying a linearly fused 4×SUMO2 protein mimicking a poly(SUMO2) chain (Extended Data Fig. 7b–d). In fact, in the presence of 4×SUMO2, NIP45-mediated SUMO conjugation was shifted from auto-modification toward SUMOylation of this substrate (Extended Data Fig. 7d). These findings are consistent with NIP45 acting as a specialized SUMO E3 ligase with SUMO chain extension activity. However, unlike RanBP2, NIP45 did not stimulate UBC9-mediated modification of a SUMOylation consensus motif (Extended Data Fig. 7a). This suggests that NIP45 might have a narrow substrate preference, perhaps restricted to its native interaction partners, in keeping with the small range of proteins displaying NIP45-dependent SUMOylation (Fig. 5f). Consistent with this idea, we found that NIP45 and SLX4 interact in cells (Fig. 5h and Extended Data Fig. 7e).

SLX4-associated nuclease activities have been shown to peak around the onset of mitosis[45] and should be considered as potential candidate effectors of NIP45-dependent catenane processing before mitosis, particularly considering the lack of other known nucleases among the NIP45-regulated SUMOylation targets identified in our proteomic analysis (Supplementary Data 7). Supporting this notion, we found that, like NIP45-KO or SUMOi treatment, depletion of SLX4, but not EME1, impaired ICRF-193-induced DSB formation, as evidenced by decreased levels of γH2AX foci and ATM-dependent CHK2 and KAP1 phosphorylation (Fig. 5i,j and Extended Data Fig. 6e). However, knockdown of SLX4 reduced ICRF-193-induced γH2AX foci formation to a lesser extent than NIP45 KO, suggesting that NIP45-dependent DNA catenane processing might be mediated by both SLX4-dependent and -independent mechanisms (Fig. 5j). In line with this proposal, loss of SLX4 did not hypersensitize cells to SUMOi treatment, and no known nucleases displayed an SL relationship with SUMOi in our CRISPR screens (Extended Data Fig. 7f and Supplementary Data 1 and 2). Importantly, however, we observed no additive impact of NIP45 KO and SLX4 depletion in reducing levels of ICRF-193-induced γH2AX foci (Fig. 5j), suggesting that NIP45 and SLX4 function in a joint pathway for converting catenanes into DSBs. Collectively, these data suggest that NIP45-dependent SUMOylation of the SLX4 multi-nuclease complex (and most likely additional factors) facilitates nucleolytic resolution of catenated DNA structures before mitotic entry to mitigate formation of UFBs and their potential for undermining chromosome segregation fidelity and cell fitness. This may contribute to the synthetic lethal interaction between RMI1 and SLX4 (Fig. 4b).

## Discussion

Our findings reveal strong SL relationships between SUMOylation, NIP45 and BTRR-PICH in human cells and establish that the essential role of SUMO signaling in cell proliferation entails a crucial function in counteracting the threat to faithful chromosome segregation posed by toxic DNA catenanes. Collectively, our data suggest a model in which SUMO acts together with NIP45 to effectuate a previously unrecognized interphase response that nucleolytically resolves catenated DNA structures before mitotic entry (Fig. 5k). The resulting DSBs trigger ATM/ATR-dependent checkpoint signaling and G2 arrest, thereby limiting the number of DNA entanglements that persist into mitosis and give rise to UFBs. This SUMO- and NIP45-dependent pathway, which is dispensable for the canonical DNA damage-induced G2 checkpoint, may be an important component of the TOP2-dependent 'decatenation checkpoint', the molecular basis of which has remained enigmatic despite the known requirement for ATM/ATR activity for its functionality[37]. We reveal that the role of NIP45 in this response relies on its ability to promote specific SUMOylation events by acting as a specialized SUMO E3 ligase via UBC9 binding through the SLD2 domain, and we

provide evidence that the SLX4 multi-nuclease scaffold constitutes one important target of NIP45-dependent SUMOylation in this pathway, the knockdown of which partially recapitulates the impairment of catenane conversion into DSBs caused by NIP45 loss. It is conceivable that SUMO modification of SLX4 could alter the functional interplay with one or more of its numerous binding partners, which include several nucleases, to facilitate nucleolytic processing of catenated DNA structures acted on by the SUMO–NIP45 pathway. Technical limitations imposed by the large size of human SLX4 precluded us from establishing directly whether, like SLX4 itself, SUMO-dependent modification of SLX4 is important for catenane cleavage into DSBs before mitosis. Moreover, our data suggest that SLX4-independent mechanisms also contribute to this process. Thus, delineating the precise mechanistic basis of catenane conversion into DSBs by SLX4-dependent and -independent effectors via NIP45-regulated SUMOylation remains an important but challenging task for future studies.

Although deliberate SUMO- and NIP45-mediated formation of DSBs before mitosis could seem counterintuitive, the conversion of DNA catenanes into DSBs accompanied by cell-cycle arrest in G2 might be critical for avoiding more severe threats to chromosome segregation and cell-division fidelity posed by catenated DNA structures. We propose that this seemingly reckless action is driven by the fact that catenanes represent 'undamaged' DNA structures, which in the absence of processing may escape detection by interphase cell-cycle checkpoints. Indeed, unlike genotoxic insults such as DSBs, which at least in some cases can be carried over to and resolved in daughter cells without gross implications for mitotic fidelity[46–48], failure to properly resolve DNA catenanes could have catastrophic consequences for chromosome segregation and cytokinesis. The SL relationships between SUMO, NIP45 and BTRR-PICH, accompanied by near-complete binucleation rates in unstressed cells, strongly suggest that the DNA catenanes acted on by the SUMO–NIP45 pathway form in virtually every cell cycle, and we consider it likely that they may correspond to DNA entanglements which, for reasons that are not yet clear, fail to be resolved by TOP2. The importance of this SUMO- and NIP45-driven pathway is consistent in principle with the conservation of NIP45 orthologs, in particular their SLDs, throughout eukaryotic evolution. Notably, however, the functions of NIP45 orthologs appear to differ between species, as Rad60 is essential in *S. pombe* whereas *S. cerevisiae* Esc2 is not required for viability, similar to our findings for human NIP45 (refs. 22,26,49). Although Esc2 and Rad60 have been implicated in several aspects of genome stability, including DSB repair and telomere maintenance[22–26], the function of human NIP45 has remained poorly defined, and the key role of NIP45 in dsDNA catenane resolution in human cells that we discovered in the present study is clearly distinct from previously reported functions of its yeast orthologs.

Our model offers a rationale for the synthetic lethality relationships between SUMO signaling, NIP45 and BTRR-PICH: in the absence of NIP45, the KO of which has no discernible impact on cell proliferation, the moderately increased level of UFBs may be effectively managed by the BTRR-PICH-dependent resolution pathway before chromosome segregation and cytokinesis. Likewise, the action of the NIP45- and SUMO-driven catenane resolution pathway operating before anaphase may enable cells to keep UFB levels below a critical threshold for binucleation in the absence of BTRR-PICH function. However, when both the NIP45 and the BTRR-PICH pathways are functionally inactivated, cells go through mitosis with elevated levels of UFBs but are unable to resolve these structures, leading to highly penetrant binucleation that undermines continued proliferation. The synthetic lethal interaction between SUMO signaling and NIP45, coupled with the notion that complete inhibition of SUMOylation phenocopies the impact of combined loss of NIP45 and BTRR-PICH function, suggests that SUMOylation may also be critical for UFB resolution via the BTRR-PICH pathway. Indeed, both BLM and PICH are known SUMOylation substrates and PICH contains SUMO-interacting motifs that target it to

mitotic chromosomes[50–52]. In this way, SUMO signaling may orchestrate complementary NIP45- and BTRR-PICH-driven catenane resolution pathways operating before and during mitosis, respectively, which together are essential for the removal of DNA entanglements that otherwise subvert chromosome segregation and cytokinesis. The genome-wide insights into genetic vulnerabilities to SUMOylation impairment and their mechanistic underpinnings reported in the present study not only shed light on the essential functions of SUMO signaling in cell proliferation but will also be important to consider in precision strategies involving pharmacological targeting of the SUMOylation machinery, which has emerged as a promising approach in cancer therapy[13].

## Online content

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

## Methods

### Cell culture

HeLa (catalog no. CCL-2), HEK293T/18 (catalog no. CRL-11268) and RPE1-hTERT (catalog no. CRL-4000) cells were obtained from American Type Cell Culture and cultured in Dulbecco's modified Eagle's medium (DMEM) GlutaMax containing 10% fetal bovine serum (FBS; Gibco) and 100 U ml$^{-1}$ of penicillin–streptomycin (Gibco). RPE1-hTERT cells with KO of puromycin acetyltransferase (RPE1-hTERT PuroS)[53] (kind gift from A. J. Holland) and RPE1-hTERT BLM−KO[17] and parental control cells (kind gifts from A. Blackford) were cultured in 50:50 DMEM GlutaMax:Ham's F-12 (Gibco) containing 10% FBS and 100 U ml$^{-1}$ of penicillin–streptomycin. RPE1-hTERT p53-KO FLAG-Cas9 cells[54] (kind gift from D. Durocher) were cultured in DMEM GlutaMax containing 10% FBS, 100 U ml$^{-1}$ of penicillin–streptomycin and 2 µg ml$^{-1}$ of Blasticidin S (Invivogen). U2OS Flp-In T-REx cells (kind gift from H. Piwnica-Worms) were cultured in DMEM GlutaMax containing 10% FBS, 100 U ml$^{-1}$ of penicillin–streptomycin, 0.1 mg ml$^{-1}$ of Zeocin (Invitrogen) and 5 µg l$^{-1}$ of Blasticidin S. U2OS Flp-In T-REx cells inducibly expressing GFP-SLX4 and parental control cells were a kind gift from J. Rouse. HeLa/His$_{10}$-SUMO2-IRES-GFP cells[55] were cultured in DMEM GlutaMax containing 10% FBS and 100 U ml$^{-1}$ of penicillin–streptomycin. KO cell lines were generated by transfection of parental cells with pX459/Cas9-(pSpCas9(BB)-2A-Puro)[56] containing sgRNAs targeting *NIP45* (5′-CTCGTCCGCGGCACCGCGAG-3′) or *RMI1* (5′-GGGTGGAGAATACAAATCCA-3′) using Lipofectamine 3000 (Invitrogen) according to the manufacturer's protocol. After 24 h of incubation, transfected cells were selected briefly (48 h) with puromycin (1 µM for HeLa and U2OS Flp-In T-REx, 2 µM for RPE1-hTERT PuroS and 30 µM for RPE1-hTERT p53-KO FLAG-Cas9) and plated sparsely. Single colonies were screened by western blotting and immunofluorescence for homogeneous KO and lack of stable Cas9 integration. All cells were cultured in humidified incubators at 37 °C with 5% CO$_2$ and were regularly tested for *Mycoplasma* infection and shown to be negative. The cell lines were not authenticated.

ML-792 (SUMOi, MedKoo), TAK-981 (Selleck Chemicals), ICRF-193 (Merck), MK1775 (WEE1i; Selleck Chemicals), nocodazole (Noco; Sigma-Aldrich), methyl methanesulfonate (MMS; Sigma-Aldrich) and mitomycin C (MMC; Sigma-Aldrich) were added to the growth medium at the doses indicated in the figures and figure legends. Unless otherwise indicated, the following doses of other compounds were used: aphidicolin (APH; 2 µM, Merck), KU55933 (ATMi; 10 µM, Toris Bioscience), AZ20 (ATRi; 1 µM, Merck), hydroxyurea (HU; 10 mM, Sigma-Aldrich), etoposide (ETP; 20 µM, Merck) and camptothecin (CPT; 2 µM, AH Diagnostics).

### Generation of stable U2OS Flp-In T-REx cell lines

U2OS Flp-In T-REx cells were co-transfected with pOG44 (Invitrogen) and a pcDNA5/FRT/TO plasmid of interest (1:9 ratio) using the Fugene 6 transfection kit (Promega) according to the manufacturer's protocol. Then, 48 h after transfection, cells were selected in medium supplemented with 200 µg ml$^{-1}$ of hygromycin B (Invitrogen) and 5 µg ml$^{-1}$ of Blasticidin S. Individual clones were selected and analyzed for homogeneous expression. Transgene expression was induced by addition of 0.1 µg ml$^{-1}$ of doxycycline (Clontech).

### Plasmids

The psPAX2 plasmid (catalog no. 12260) and pMD2.G plasmid (catalog no. 12259) were purchased from Addgene. The pEVRF/NLS-GFP-L2-PCNA plasmid[57] was a kind gift from C. Cardoso (TU Darmstadt, Germany). Flag-HA-SLX4 plasmid[44] was a kind gift from P.-H. Gaillard (CRCM, France). The pcDNA5/FRT/TO/GFP−NIP45 was generated by inserting GFP with HindIII followed by full-length human *NIP45* complementary DNA (cDNA) with KpnI and NotI using the following primers: Fw_EGFP-N_HindIII, Rw_EGFP-N_HindIII, Fw_NIP45_KpnI and Rv_NIP45_NotI. The pcDNA5/FRT/TO/mCherry−NIP45 was generated

by inserting mCherry with AflII and KpnI followed by full-length human *NIP45* cDNA with KpnI and NotI using the following primers: Fw_mCherry_AflII; Rv_mCherry_KpnI and NIP45 primers as above. Constructs containing deletion of NIP45 SLD1 (ΔSLD1; amino acids 261–335) were generated using the Q5 site-directed mutagenesis kit (New England Biolabs) according to the manufacturer's protocol, using the following primers: Fw_NIP45_SLD1 and Rv_NIP45_SLD1. Constructs with a deletion of NIP45 SLD2 (ΔSLD2; amino acids 344–419) were cloned using the following primers: Fw_NIP45_KpnI (as above) and Rv_NIP45_SLD2_NotI. Constructs with a deletion of part of the NIP45 N-terminal (ΔN1; amino acids 208–419) were generated using the following primers: Fw_NIP45_208-419_KpnI and Rv_NIP45_NotI (as above). Constructs with a deletion of the entire NIP45 N-terminal (ΔN2; amino acids 261–419) were generated using the following primers: Fw_NIP45_261-419_KpnI and Rv_NIP45_NotI (as above). Constructs with the NIP45 D394R mutation (SLD2*) were generated by site-directed mutagenesis using the following primers: Fw_NIP45_D394R and Rv_NIP45_D394R. A complete list of primer sequences used in the present study is provided in Supplementary Table 1. All constructs were verified by full sequencing across the inserts.

### Antibodies

A complete list of commercially available antibodies used in the present study is provided in Supplementary Table 2. The following customized antibodies were used: NIP45 (sheep polyclonal, raised against full-length human NIP45; western blotting: 1:1,000; immunofluorescence: 1:1,000), RMI1 (mouse monoclonal; western blotting: 1:1,000), PICH (guinea-pig polyclonal; western blotting: 1:200; immunofluorescence: 1:500) and SLX4 (rabbit polyclonal, gift from John Rouse, University of Dundee; western blotting: 1:1,000).

### siRNAs

siRNA transfections were performed using Lipofectamine RNAiMAX (Invitrogen) according to the manufacturer's instructions. A complete list of siRNA sequences used in the present study is provided in Supplementary Table 3. BLM siRNAs were used as an equimolar mixture of four siRNAs (siBLM nos. 1–4) at a final concentration of 50 nM and SLX4 siRNAs were used as an equimolar mixture of two siRNAs (siSLX4 nos. 1 and 2) at a final concentration of 50 nM.

### Genome-scale CRISPR−Cas9 screens

Toronto human KO pooled library (TKOv3) (Addgene, catalog nos. 90294 and 125517) was a gift from J. Moffat[14,58]. The TKOv3 library contains 71,090 sgRNA sequences targeting 18,053 human protein-coding genes with a modal number of four sgRNAs per gene. Viral particles of the LCV2::TKOv3 and pLCKO2::TKOv3 sgRNA libraries were produced as previously described[14]. Briefly, HEK293T/18 cells were seeded 24 h before transfection with a mix of TKOv3-pooled plasmid library, psPAX2 plasmid and pMD2.G plasmid using Lipofectamine 3000 according to the manufacturer's instructions. Medium was changed 6 h post-transduction to DMEM GlutaMax containing 10% FBS, 100 U ml$^{-1}$ of penicillin–streptomycin and 1% bovine serum albumin (BSA; Sigma-Aldrich). Medium containing viral particles was collected and filtered through a 0.45-µm filter 48 h post-transfection and stored at −80 °C. RPE1-hTERT p53-KO FLAG-Cas9, NIP45−KO and RMI1−KO cell lines were transduced with the pLCKO2::TKOv3 library at a low multiplicity of infection (MOI; 0.2–0.4) with a coverage of >350-fold sgRNA representation, which was maintained throughout the screens at each cell passage point. HeLa parental and NIP45−KO cell lines were transduced with the LCV2::TKOv3 library at a low MOI (-0.25) and a coverage of >250-fold sgRNA representation, which was maintained throughout the screen at each cell passage point. For RPE1-hTERT p53-KO FLAG-Cas9 cell lines, cells were selected for 24 h with 25 µg ml$^{-1}$ of puromycin 1 d after transduction and then trypsinized and reseeded in the same plates while maintaining puromycin selection for another

24 h. For HeLa cell lines, cells were selected for 48 h with 1 µg ml⁻¹ of puromycin 1 d after transduction. Then, 3 d after transduction, which was considered the initial time point (t0), cells were pooled and passaged whereas cell pellets of two replicates of 3 × 10⁷ cells were frozen for downstream processing. Cells were passaged after another 3 d and 9 d after transduction (t6) cells were split into technical duplicates. For synthetic lethality screens, cells were passaged another 12 d (t6–t18) with passaging every 3 d. For SUMOi sensitivity screens, cells were passaged every 3 d (t6, t9, t12 and t15) in medium with or without a low dose of SUMOi (56 nM for HeLa cells; 125 nM for RPE1-hTERT p53-KO FLAG-Cas9 cells) equivalent to predetermined LD$_{20}$ concentrations in uninfected cells. At the final time point (t18) cell pellets from 3 × 10⁷ cells were frozen from each replicate.

Genomic DNA from cells collected at t0 and t18 was isolated as previously described[59]. Briefly, cell pellets from 3 × 10⁷ cells were lysed overnight at 55 °C in 6 ml of NK buffer (50 mM Tris, pH 8.0, 50 mM EDTA and 1% sodium dodecylsulfate (SDS)) containing 0.1 mg ml⁻¹ of Proteinase K (Merck), and then incubated for 30 min at 37 °C with RNase A (QIAGEN) at a final concentration of 50 µM. Samples were cooled on ice before addition of 2 ml of pre-chilled 7.5 M ammonium acetate (Sigma-Aldrich) to precipitate proteins. Samples were then vortexed and centrifuged at ≥4,000g for 10 min at 4 °C. Supernatant was mixed with 6 ml of isopropanol and centrifuged at ≥4,000g for 10 min at 4 °C. Genomic DNA precipitate was washed once in 70% ethanol, air dried and resuspended in 0.1× TE buffer (1 mM Tris, pH 8.0 and 0.1 mM EDTA). The region of genomic integration containing sgRNA sequences was amplified by PCR using Q5 Mastermix Next Ultra II (New England Biolabs) with the following primers: pLCKO2_forward and pLCKO2_reverse or LCV2_forward and LCV2_reverse (Supplementary Table 1). This was followed by a second PCR reaction containing i5 and i7 multiplexing barcodes and final gel-purified products were sequenced on Illumina NextSeq500. Fastq files were generated using bcl2fastq v.2.19.1 and reads were trimmed to 20 bp using cutadapt 1.18, removing a variable number of basepairs at the start and end depending on the size of the primer stagger. MAGeCK 0.5.8 (ref. 60) was used to assign the trimmed reads to the guides in the TKOv3 library and create the count matrix. To identify genes required for cell survival in the presence of SUMOi, gene scores (NormZ values) were estimated from the count matrix using the drugZ algorithm[61], applying a NormZ value of <−3 as a cut-off for significant hits.

To identify synthetic lethal genes in NIP45–KO and RMI1–KO backgrounds, we compared sgRNA depletion in WT and KO backgrounds using the BAGEL (Bayesian Analysis of Gene EssentiaLity) algorithm (t0 versus t18)[62,63]. The delta BAGEL factor (delta_BF) was calculated for each gene by subtracting the BAGEL factor (BF) of KO cells from the BF of parental cells. Synthetic lethal genes were defined as genes with both delta_BF > 15 and KO BAGEL factor >15. To assess data quality of the CRISPR screens, we generated precision-recall curves through the BAGEL.py 'pr' function[62] using the core essential (CEGv.2.txt) and nonessential (NEGv.1.txt) gene lists from https://github.com/hart-lab/bagel, comparing t0 with t18 for mock-treated cells.

### Whole-cell extracts, immunoprecipitation and western blotting

For whole-cell extracts, cells were lysed for 15 min in ice-cold RIPA buffer (25 mM Tris-HCl, pH 7.5, 150 mM NaCl, 1 mM EDTA, 1% NP-40, 0.5% sodium deoxycholate, 0.1% SDS and 1 mM dithiothreitol (DTT)) containing 1 mM NaF, 10 mM N-ethylmaleimide, 10 mM β-glycerophosphate, 0.1 mM vanadate and complete EDTA-free protease inhibitor cocktail (Roche), and sonicated for 20 s. Cell lysates were cleared by centrifugation at 16,100g and 4 °C. For detection of TOP2A, cells were collected by scraping in TD buffer (20 mM Tris-HCl, pH 7.5, 100 mM NaCl, 20 mM KCl and 0.5 mM Na$_2$HPO$_4$), and cell pellets lysed for 10 min in ice-cold buffer A (20 mM Tris-HCl, pH 7.5, 100 mM NaCl, 50 mM KCl, 0.1 mM EDTA, 0.1 mM phenylmethylsulfonyl fluoride, 10% glycerol, 0.2% NP-40 and

0.1% Triton X-100). Nuclei were pelleted by centrifugation at 1,000g and 4 °C, lysed in buffer A containing 1% SDS and sonicated for 15 s. Protein concentration was determined using the Pierce BCA protein assay kit (Thermo Fisher Scientific) before addition of 1× Laemmli SDS sample buffer (final concentration: 50 mM Tris, pH 6.8, 10% glycerol, 100 mM DTT, 2% SDS and 0.1% Bromophenol Blue) and boiling for 5 min.

For GFP-trap pulldowns, U2OS Flp-In T-REx NIP45–KO/GFP–NIP45 or U2OS Flp-In T-Rex GFP-SLX4 cell lines were induced with 0.1 µM doxycycline for 24 h and cell pellets collected and lysed for 15 min in ice-cold low-salt buffer (50 mM Tris-HCl, pH 7.5, 50 mM NaCl, 1 mM EDTA, 1 mM DDT and 1% NP-40) containing 1 mM NaF, 10 mM N-ethylmaleimide, 10 mM β-glycerophosphate, 0.1 mM vanadate and complete EDTA-free protease inhibitor cocktail. For Flag-HA-SLX4 and mCherry–NIP45 co-expression, cells transfected using Lipofectamine 3000 according to the manufacturer´s instructions were collected 24 h post-transfection and lysed for 15 min in ice-cold low-salt buffer. Cell lysates were cleared by centrifugation at 16,100g and 4 °C. Protein concentration was determined using the Pierce BCA protein assay kit and equalized with lysis buffer before addition to 30 µl of GFP-trap bead/slurry pre-washed twice in lysis buffer. After incubation for 45–120 min at 4 °C, the beads were washed 3× in lysis buffer and proteins eluted with 2× Laemmli SDS sample buffer with boiling for 5 min.

For NIP45 immunoprecipitation, cells were lysed for 15 min in ice-cold EBC buffer (50 mM Tris-HCl, pH 7.5, 150 mM NaCl, 1 mM EDTA and 0.5% NP-40) containing 1 mM NaF, 10 mM N-ethylmaleimide, 10 mM β-glycerophosphate, 0.1 mM vanadate and complete EDTA-free protease inhibitor cocktail. Cell lysate was sonicated for 20 s and cleared by centrifugation at 16,100g and 4 °C. Protein concentration was determined using the Pierce BCA protein assay kit and equalized with lysis buffer. Protein G-coupled beads were pre-incubated for 16 h with 3 µg of sheep immunoglobulin (Ig)G or sheep anti-NIP45 antibody, washed twice in lysis buffer and incubated with cell lysate at 4 °C for 4 h. Beads were washed 3× in lysis buffer and proteins eluted with 2× Laemmli SDS sample buffer with boiling for 5 min.

Whole-cell extracts and immunoprecipitations were analyzed by SDS–polyacrylamide gel electrophoresis on NuPage Bis–Tris 4–12% protein gels (Invitrogen), and proteins were transferred to poly(vinylidene fluoride) membranes (Immobilon-FL, Merck). For western blotting with phospho-specific antibodies, membranes were blocked in 5% BSA in TBS-T (Tris-buffered saline with Tween 20), incubated with primary antibody in 5% BSA TBS-T overnight at 4 °C, washed in TBS-T, incubated with secondary antibody in 5% BSA in TBS-T for 1 h and washed again in TBS-T. For western blotting with all other antibodies, membranes were blocked in 5% skimmed-milk PBS-T (phosphate-buffered saline with Tween 20), incubated with primary antibody in 2% skimmed-milk PBS-T overnight at 4 °C, washed in PBS-T, incubated with secondary antibody in 2% skimmed-milk PBS-T for 1 h and washed again in PBS-T. Membranes were imaged with the Odyssey CLx (LI-COR) using ImageStudio (v.3.1.4, LI-COR) or incubated with ECL reagent and imaged on an ImageQuant LAS4000 (Cytiva) using ImageQuant LAS4000 software (v.1.2, GE healthcare).

### Immunofluorescence microscopy

For analysis of UFBs, asynchronously growing cells were seeded on sterile glass coverslips at 20% confluency. The next day, the medium was replaced by fresh medium containing SUMOi (ML-792, 50 nM), aphidicolin (0.4 µM), ICRF-193 (0.1 µM) or dimethylsulfoxide, depending on the experiment. Cells were further incubated for 16–24 h and fixed in co-extraction buffer (20 mM 1,4-piperazinediethanesulfonic acid, pH 6.8, 1 mM MgCl$_2$, 10 mM (ethylenebis(oxonitrilo))tetra-acetate, 4% formaldehyde and 0.2% Triton X-100) for 15 min at room temperature (RT). The buffer was discarded, and the cells were rinsed immediately with PBS. The cells were washed further with PBS for 5 min and this was repeated 3×. Cells were permeabilized with PBSAT buffer (3% BSA and 0.5% Triton X-100 in PBS) overnight at 4 °C. The cells were then washed

3× with PBS for 5 min. PICH-positive UFBs were stained using PICH antibody and goat anti-guinea-pig IgG Alexa Fluor-488 (Invitrogen, diluted 1:1,000 in PBSAT buffer). After incubation with antibodies, cells were stained with DAPI and mounted with DAPI-free Vectashield mounting medium (Vector Laboratories). Images were acquired using an Olympus BX63 microscope and processed in Fiji. For each experiment, quantification of PICH-positive UFBs was performed in at least 80 late anaphase cells (anaphase B) per condition per replicate in 3 independent biological replicates.

To determine the mitotic index after treatment with ICRF-193, cells were seeded on sterile glass coverslips, allowed to adhere and then treated for 16 h with nocodazole (0.5 mM for U2OS Flp-In T-REx; 1 mM for RPE1 PuroS) in the presence or absence of the indicated drugs. Cells were carefully washed once in PBS and fixed in formalin buffer (VWR) for 15 min at RT. Cells were permeabilized with PBS containing 0.2% Triton X-100 for 5 min and blocked with PBS containing 3% BSA for 1 h before staining for 2 h at RT with phospho-MPM2-Cy5 conjugate (Merck, 1:500) and DAPI. The mitotic index was determined in each condition (number of phospho-MPM2-positive cells/total number of cells) and normalized to nocodazole treatment alone to obtain the relative mitotic index. To determine cell-cycle distribution, asynchronously growing cells were incubated for 20 min with 10 μM 5-ethynyl-2′-deoxyuridine (EdU; Thermo Fisher Scientific), fixed and permeabilized as described above. Nascent DNA was labeled with Click-iT Plus EdU Alexa Fluor-647 Imaging Kit (Thermo Fisher Scientific) according to the manufacturer's instructions, followed by staining with DAPI. For analysis of γH2AX and 53BP1 foci, asynchronously growing cells were seeded on sterile glass coverslips. After treatment with the indicated drugs, cells were washed once in PBS and fixed in formalin buffer for 15 min at RT. Cells were permeabilized with PBS containing 0.2% Triton X-100 for 5 min and blocked with PBS containing 3% BSA for 1 h before staining for 2 h at RT with γH2AX (1:500) or 53BP1 (1:500) antibodies, washed 3× in PBS and stained for 1 h at RT with secondary antibody and DAPI. Quantitative image-based cytometry was performed as described previously[64]. In brief, images were acquired with an Olympus IX-81 wide-field microscope equipped with an MT20 Illumination system and a digital monochrome Hamamatsu C9100 CCD camera. Olympus UPLSAPO ×10/0.4 numerical aperture (NA) and ×20/0.75 NA objectives were used. Automated, unbiased image analysis was carried out with the ScanR analysis software (v.2.8.1). Data were exported and processed using Spotfire software (v.10.5.0; Tibco).

## Cell growth assays

The sulforhodamine B (SRB) colorimetric assay[65] was used to quantify cell growth. HeLa, U2OS Flp-In T-REx and RPE1 PuroS cells transfected with siRNAs or left untreated were seeded (700 cells for HeLa, 500 cells for U2OS Flp-In T-REx and 150 cells for RPE1 PuroS) in 24-well plates in medium containing the indicated drug doses for 3 d. The medium was then changed and cells were grown for an additional 9 d. For treatment with ICRF-193, CPT, ETP, MMS, MMC, APH, HU and IR, cells were seeded in 24-well plates 24 h before treatment with drugs at the indicated doses and duration. After a total of 12 d of growth, cells were washed once in PBS and fixed with 10% (w:v) trichloroacetic acid for 30 min at 4 °C. After two washes with deionized water, cells were stained with 0.4% (w:v) SRB (Sigma-Aldrich) in 1% acetic acid for 20 min at RT. Cells were then washed 4× with 1% acetic acid and the plates left to dry overnight. Protein-bound SRB was dissolved in 10 mM Tris, pH 8.0, for 2 h at RT with shaking and absorbance (510 nm) was measured on a FLUOstar Omega (BMG Labtech) plate reader and analyzed by the accompanying MARS data analysis software.

Relative cell density was measured using an Incucyte S3 Live-Cell Analysis System. RPE1 PuroS WT and NIP45−KO cell lines were seeded in duplicate in 24-well plates ($2 \times 10^3$ cells per well) in the presence or absence of SUMOi (500 nM) and allowed to adhere for 24 h. Cells were imaged at 6-h intervals with a mean confluency determined from 16 images per well and normalized to the starting time point.

## Live-cell microscopy

Live-cell microscopy was performed using a Deltavision Elite microscope (GE Healthcare) equipped with a ×40 oil objective lens with an NA of 1.35 (GE Healthcare). Before live-cell microscopy, cells were transfected and treated with drugs or siRNAs as indicated. The day before filming cells were seeded into eight-well culture slides (Ibidi). For quantification of G2 length, cells were transfected with pEVRF/NLS-GFP-L2-PCNA the day before filming and treated with 7 mM ICRF immediately before filming. G2 length was quantified as previously described[66]. Briefly, G2 length was defined as timing from disappearance of proliferating cell nuclear antigen (PCNA) foci (end of S phase) to nuclear envelope breakdown (beginning of mitosis). During live-cell microscopy, cells were maintained at 37 °C in Leibovitz's L-15 medium (Gibco) containing 10% fetal calf serum. SoftWoRx software (GE Healthcare) was used to acquire and subsequently analyze the data. The DeltaVision Elite microscope was equipped with a CoolSNAP HQ2 camera (Photometrics).

## Neutral comet assays

DSB formation was analyzed by neutral single-cell gel electrophoresis using the CometAssay kit (Trevigen) according to the manufacturer's instructions. Images were acquired with a Leica AF6000 wide-field microscope (Leica Microsystems) equipped with HC PL APO ×20/0.7 NA objective, using standard settings. Image acquisition and analysis were carried out with Leica Application Suite X software (Leica Microsystems) and the tail moment of at least 50 cells per experiment was analyzed with the TriTek CometScore software.

## Flow cytometry

Asynchronously growing HeLa WT and NIP45−KO cells were either treated or not treated with IR (4 Gy) followed by nocodazole (150 ng ml$^{-1}$) for 4 h. Cells were collected and fixed in 70% ethanol at 4 °C and stained with phospho-MPM2 antibody (1:1,000) for 2 h at RT. Flow cytometry analysis was carried out on a 5-laser Becton Dickinson LSR Fortessa instrument using BD FACS Diva software (v.9.0) for data acquisition and FCS Express (v.7; DeNovo Software) for data analysis. Quality control was done on the instrument using the Cytometer Set-up and Tracking program and beads before analysis.

## In vitro SUMOylation assays

Conjugation assays contained 50 mM Tris-HCl, pH 7.5, 50 mM NaCl, 1 mM tris(2-caraboxyethyl) phosphine hydrochloride, 5 mM MgCl$_2$, 2 mM ATP, 110 nM SAE1/SAE2 and 1 μM UBC9. SUMO1 and SUMO2 were used at either 5 or 11 μM with 1 μM fluorescently labeled SUMO as indicated. GST-SP100(241−360), GST-NIP45, GST, RanBP2(2532-2767) and 4×SUMO2 were added at 0.5 μM. FITC-SRBD1 peptide (TFGQSALK-KIKTETYPQGQPV; obtained from peptide 2.0) was added at 3.0 μM. Reactions were incubated at 37 °C for the indicated times. Reactions were analyzed by Coomassie staining or fluorescence detection using a Typhoon (Amersham). SAE1/SAE2, UBC9, SUMO1 and SUMO2 (ref. 67), RanBP2(2532-2767)[68] and 4× SUMO2 (ref. 69) were all expressed and purified as described. GST-NIP45 and GST-SP100(241−360) were purified from bacteria as described[70]. SUMO1 and SUMO2 containing a single cysteine residue were labeled with Alexa Fluor-488 or -647 as described[71].

## Quantification and statistical analysis

All statistical analyses were performed using Prism v.9.3.0 (GraphPad Software). Statistical details including number of independent experiments (*n*), definition of significance and measurements are defined in figure legends. No statistical method was used to predetermine sample size and no data were excluded from the analyses. Samples were not

randomized and investigators were not blinded to group allocation during data collection and analysis.

## Reporting summary

Further information on research design is available in the Nature Portfolio Reporting Summary linked to this article.

## Data availability

The MS proteomics data (Supplementary Data 6 and 7) have been deposited to the ProteomeXchange Consortium[72] via the Proteomics Identifications (PRIDE) partner repository (http://www.ebi.ac.uk/pride) under accession no. PXD033739. All other data supporting the findings of the present study are available within the article and supplementary information. Source data are provided with this paper.

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

## Acknowledgements

We thank D. Durocher (Lunenfeld-Tanenbaum Research Institute, University of Toronto, Canada), A. J. Holland (Johns Hopkins University School of Medicine, MA, USA), A. Blackford (University of Oxford, UK), J. Moffat (University of Toronto, Canada), H. Piwnica-Worms (MD Anderson Cancer Center, University of Texas), C. Cardoso (Technical University (TU) of Darmstadt, Germany), P.-H. Gaillard (Aix Marseille University, France) and J. Rouse (University of Dundee, UK) for providing reagents, J. Lukas for critical reading of the manuscript and members of the Mailand laboratory for helpful discussions. We thank M. Olivieri and D. Durocher for help with implementing genome-scale CRISPR–Cas9 screening, G. de la Cruz for assistance with flow cytometry and M. Michaut and H. Neil of the Genomics Platform at Novo Nordisk Foundation Center for Protein Research and Center for Stem Cell Medicine for technical support and use of instruments. Data processing and analysis were performed using the DeiC National Life Science Supercomputer at Technical University of Denmark (www.computerome.dk). This work was supported by grants from the Novo Nordisk Foundation (grant nos. NNF14CC0001 (to M.L.N., J.N. and N.M.) and NNF18OC0030752 (to N.M.)), Independent Research Fund Denmark (grant nos. 7016-00055B and 0134-00048B (to N.M.)), Lundbeck Foundation (grant no. R223-2016-281 (to N.M.)), Nordea Foundation (to I.D.H.) and Danish National Research Foundation (grant no. DNRF-115 (to I.D.H.)).

## Author contributions

E.P.T.H. and N.M. conceived the project. E.P.T.H., T.K., I.A.-d.V., I.A.H., A.H.B., A.E.-A., R.T.H., M.L.N., J.N., I.D.H. and N.M. provided the methodology. E.P.T.H., T.K., I.A.-d.V., I.A.H., Y.W., A.H.B. and A.E.-A. carried out the investigations. N.M. wrote the original draft. All authors reviewed and edited the manuscript. R.T.H., M.L.N., J.N., I.D.H. and N.M. supervised the project. N.M. administered the project. I.D.H. and N.M. acquired the funding.

## Competing interests

The authors declare no competing interests.

## Additional information

**Extended data** is available for this paper at https://doi.org/10.1038/s41594-023-01045-0.

**Correspondence and requests for materials** should be addressed to Emil P. T. Hertz or Niels Mailand.

**Peer review information** *Nature Structural & Molecular Biology* thanks Dana Branzei, Joanna Morris and Stefan Mueller for their contribution to the peer review of this work. Peer reviewer reports are available. Primary Handling Editors: Florian Ullrich and Carolina Perdigoto, in collaboration with the *Nature Structural & Molecular Biology* team. Peer reviewer reports are available.

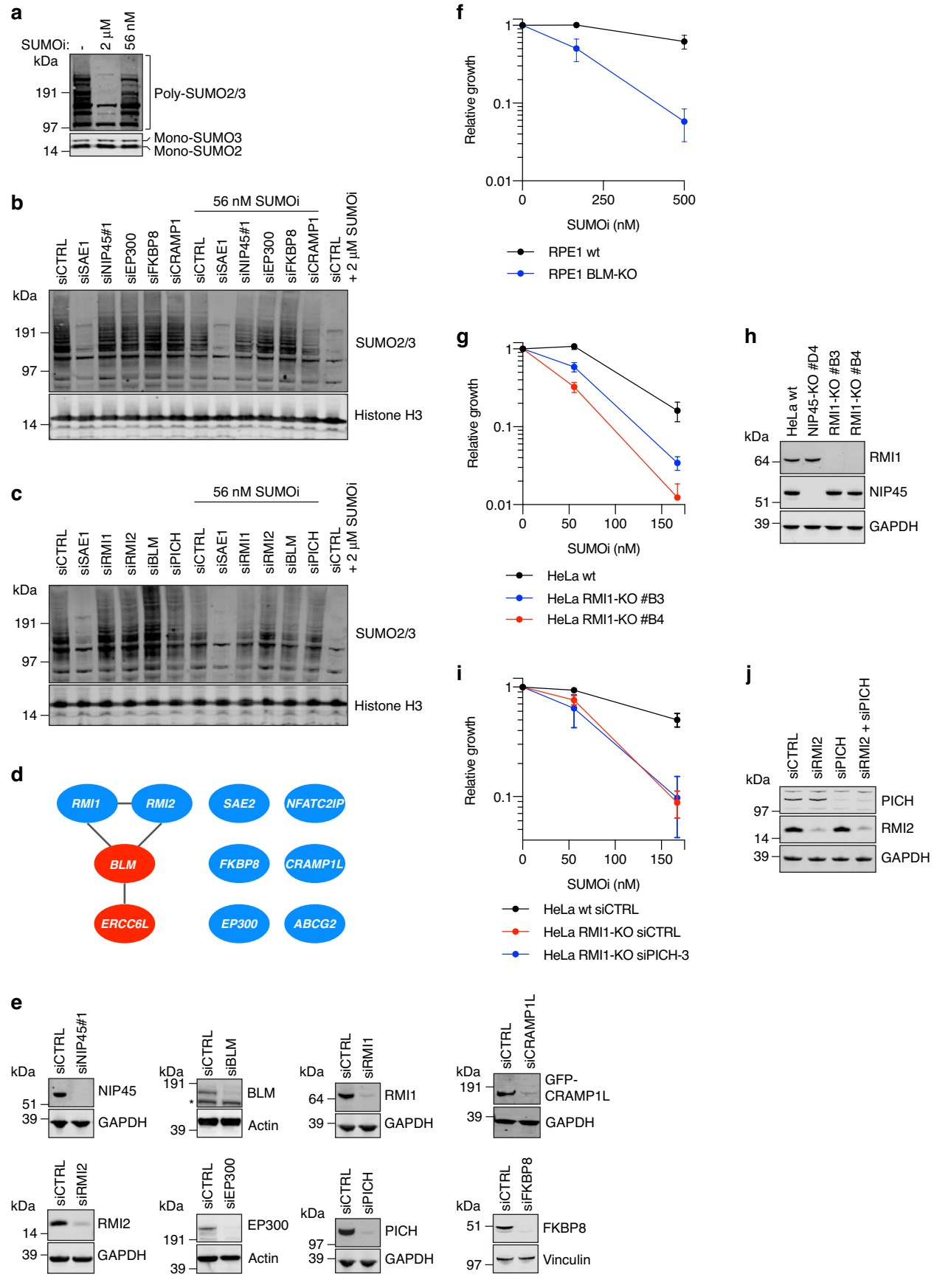

**Extended Data Fig. 1 | See next page for caption.**

**Extended Data Fig. 1 | Validation of SUMOi CRISPR-Cas9 screen hits. a.** Immunoblot analysis of whole cell lysates from HeLa cells that were left untreated or subjected to low-dose (56 nM; corresponding to the LD20) or high-dose (2 µM) SUMOi treatment for 2 h. **b.** Immunoblot analysis of whole cell lysates from HeLa cells treated with low-dose (56 nM) or high-dose (2 µM) SUMOi for 2 h following transfection with control (CTRL), SAE1, NIP45, EP300, FKBP8 or CRAMP1 siRNAs. **c.** As in (**b**), but using CTRL, SAE1, RMI1, RMI2, BLM or PICH siRNAs. **d.** Schematic representation of common hits (blue) from genome-scale CRISPR-Cas9 screens in HeLa and RPE1 cells for genes whose KO sensitizes cells to SUMOi. Genes identified in one screen only are highlighted in red. **e.** Immunoblot analysis of siRNA-mediated knockdown efficiency in whole cell lysates from HeLa cells. **f.** SRB cell growth assay using RPE1 wt and BLM-KO cells treated with indicated doses of SUMOi (mean±s.d.; *n* = 3 independent experiments). **g.** SRB cell growth assay using HeLa wt and RMI1-KO cell lines treated with indicated doses of SUMOi (mean±s.d.; *n* = 3 independent experiments). **h.** Immunoblot analysis of whole cell lysates from HeLa wt, NIP45-KO and RMI1-KO cell lines. **i.** SRB cell growth assay using HeLa wt and RMI1-KO cell lines treated with indicated siRNAs and SUMOi doses (mean±s.d.; *n* = 3 independent experiments). **j.** Immunoblot analysis of whole cell lysates from HeLa cells transfected with indicated siRNAs.

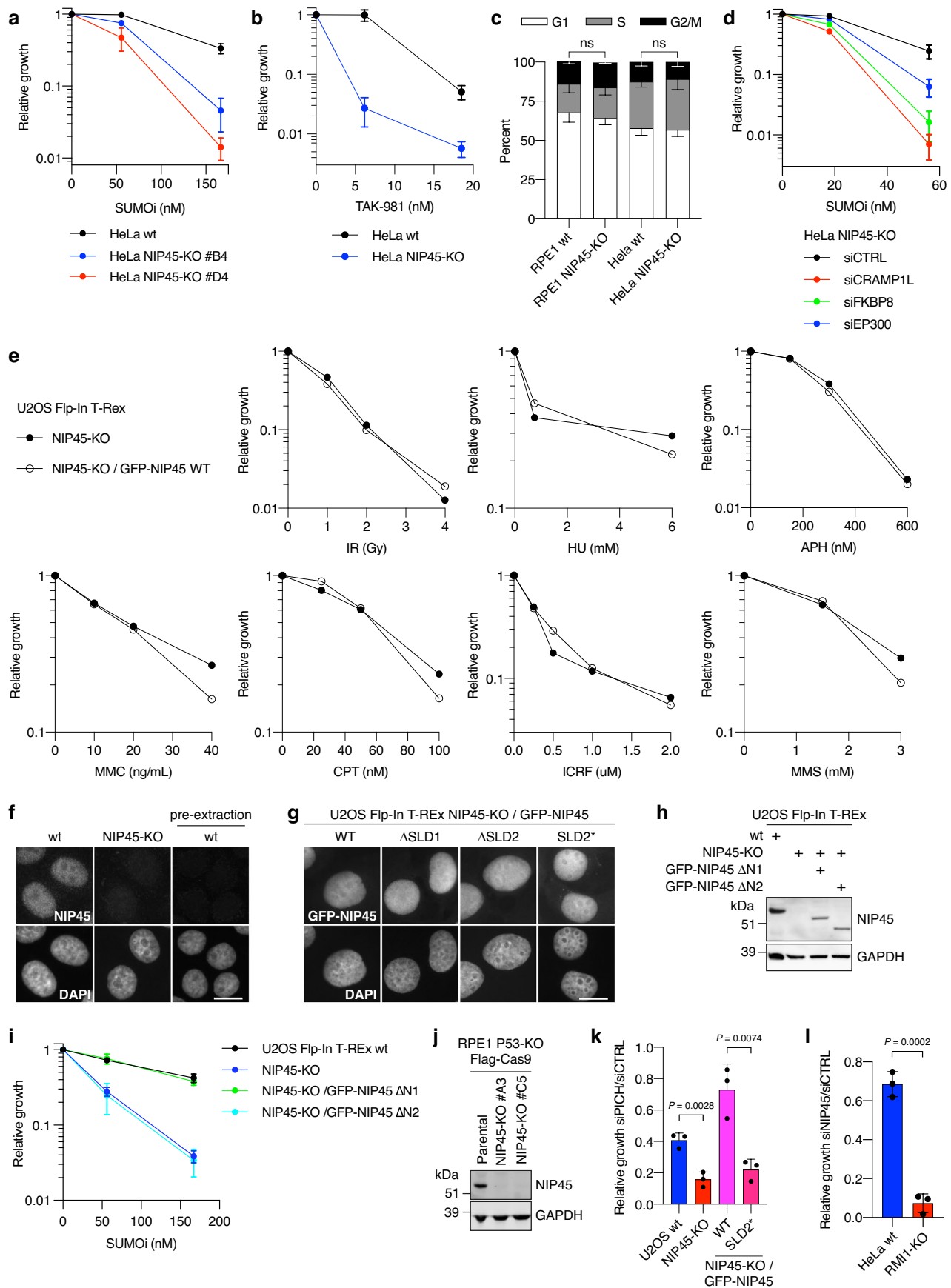

**Extended Data Fig. 2 | See next page for caption.**

**Extended Data Fig. 2 | Characterization of phenotypes associated with NIP45 deficiency and mutation. a**. SRB cell growth assay using HeLa wt and NIP45-KO cell lines treated with indicated doses of SUMOi (ML-792) (mean ± s.d.; $n$ = 3 independent experiments). **b**. As in (**a**), using indicated doses of TAK-981 (mean ± s.d.; $n$ = 3 independent experiments). **c**. Cell cycle analysis of asynchronously growing RPE1 wt and NIP45-KO, and HeLa wt and NIP45-KO cell lines using EdU incorporation and DAPI intensity to distinguish cell cycle phases (mean±s.d.; $n$ = 3 independent experiments; unpaired two-tailed $t$-test). **d**. SRB cell growth assay using HeLa NIP45-KO cells treated with indicated SUMOi doses following transfection with siRNAs (mean±s.d.; $n$ = 3 independent experiments). **e**. SRB cell growth assay using U2OS Flp-In T-REx NIP45-KO and U2OS Flp-In T-REx NIP45-KO/GFP-NIP45 FL cell lines treated with compounds at indicated doses 24 h post seeding (ICRF-193 and aphidicolin, continuous treatment; CPT, MMC and HU, 24 h treatment; MMS, 1 h treatment) (mean of $n$ = 2 technical replicates). **f**. Representative images of HeLa wt and NIP45-KO cells immunostained with NIP45 antibody with or without Triton X-100 pre-extraction to remove soluble proteins. Scale bar, 10 µM. **g**. Representative images of U2OS Flp-In T-REx NIP45-KO/GFP-NIP45 cell lines immunostained with GFP antibody. Scale bar, 10 µM. **h**. Immunoblot analysis of NIP45 protein levels in whole cell lysates from U2OS Flp-In T-Rex wt and NIP45-KO cell lines inducibly expressing indicated GFP-NIP45 variants. **i**. SRB cell growth assay using U2OS Flp-In T-Rex wt and NIP45-KO cell lines stably expressing indicated GFP-NIP45 mutants that were treated with indicated SUMOi doses (mean±s.d.; $n$ = 3 independent experiments). **j**. Immunoblot analysis of whole cell lysates from RPE1 p53-KO FLAG-Cas9 parental or NIP45-KO cell lines used for genome-scale CRISPR-Cas9 screens for synthetic lethality. **k**. SRB cell growth assay using U2OS Flp-In T-REx wt and NIP45-KO cell lines stably expressing GFP-NIP45 WT or SLD2*, comparing the impact of PICH to control (CTRL) siRNAs (mean±s.d.; $n$ = 3 independent experiments; unpaired two-tailed $t$-test). **l**. SRB cell growth assay using HeLa wt and RMI1-KO cell lines, comparing the impact of NIP45 and control (CTRL) siRNAs (mean±s.d.; $n$ = 3 independent experiments; unpaired two-tailed $t$-test). Data information: Data are representative of three (a) and two (f,j) independent experiments with similar outcome.

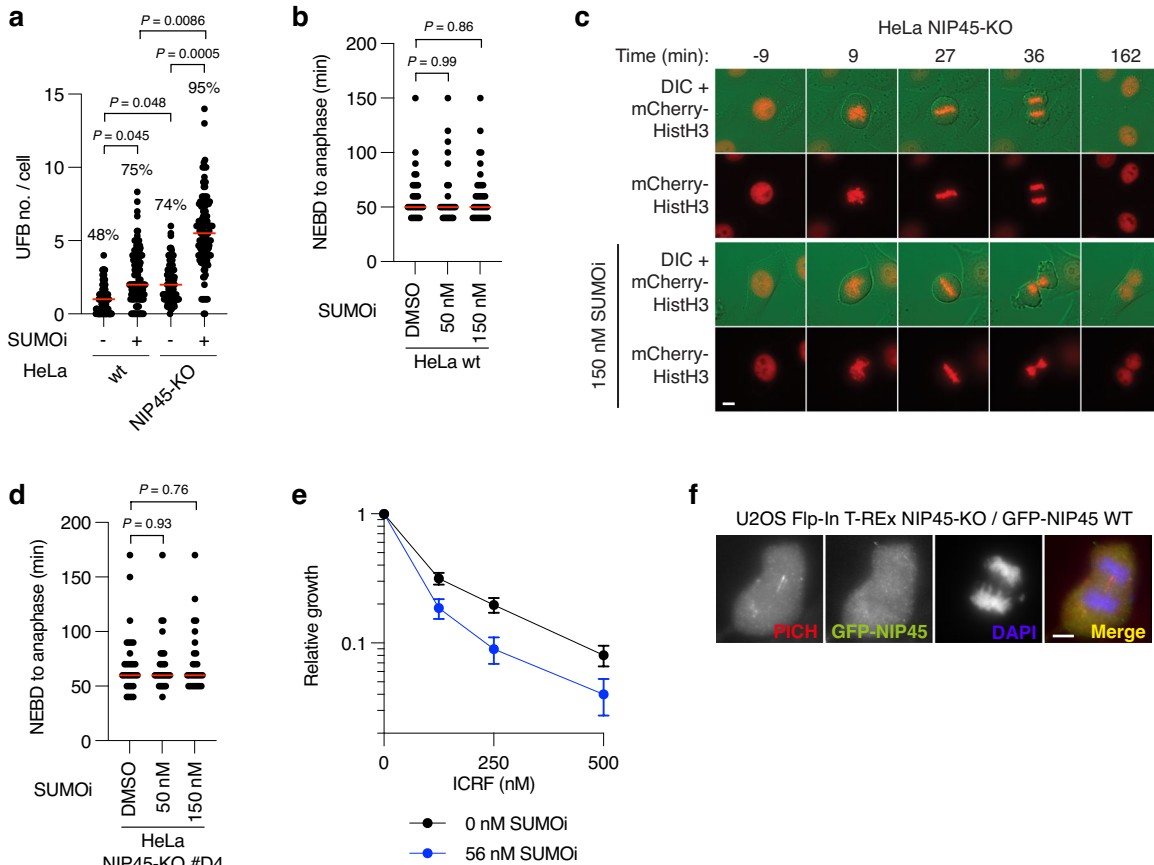

**Extended Data Fig. 3 | Impact of NIP45 KO and SUMOi treatment on mitotic progression and abnormalities. a**. Quantification of UFBs following treatment with SUMOi (50 nM) for 24 h in HeLa wt and NIP45-KO cells (all data points are shown; red bars, median; $n$ = 3 independent experiments; at least 80 cells scored per condition per independent experiment; unpaired two-tailed $t$-test). Percentage above bars indicates fraction of cells containing at least one UFB (UFB-positive cells). **b**. Quantification of NEBD to anaphase onset duration in HeLa cells wt treated with indicated SUMOi doses using live cell imaging (red bars, median; representative experiment of $n$ = 3 independent experiments; at least 27 cells were scored per condition per replicate; unpaired two-tailed Mann-Whitney test). **c**. Representative live cell microscopy images of HeLa NIP45-KO cells transiently expressing mCherry-Histone H3 following treatment with or without SUMOi (150 nM) for 48 h. Indicated times are relative to nuclear envelope breakdown (NEBD). DIC, differential interference contrast. Scale bar: 10 μm. Data are representative of 3 independent experiments with similar outcome. **d**. Quantification of NEBD to anaphase onset duration in HeLa NIP45-KO cells treated with indicated doses of SUMOi using live cell imaging (red bars, median; representative experiment of $n$ = 3 independent experiments; at least 26 cells were scored per condition per replicate; unpaired two-tailed Mann-Whitney test). **e**. SRB cell growth assay using HeLa cells treated with indicated SUMOi and ICRF-193 doses (mean±s.d.; $n$ = 3 independent experiments). **f**. Representative immunofluorescence images of U2OS Flp-In T-REx NIP45-KO/GFP-NIP45 cells in late anaphase immunostained with PICH antibody. Scale bar, 5 μM.

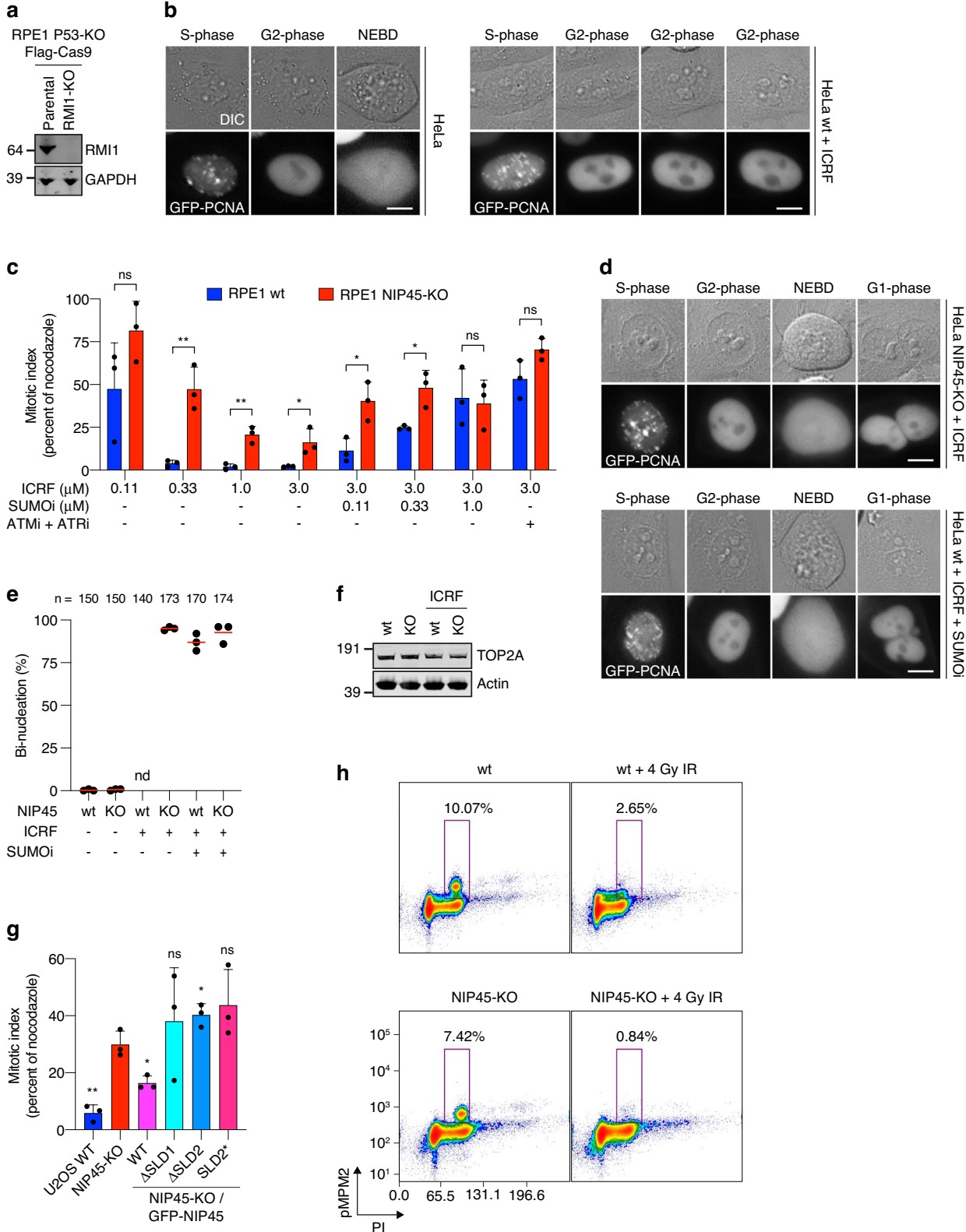

**Extended Data Fig. 4 | See next page for caption.**

**Extended Data Fig. 4 | Impact of NIP45 KO and SUMOi treatment on G2 checkpoints. a**. Immunoblot analysis of whole cell lysates from RPE1 p53-KO FLAG-Cas9 parental cells and the RMI1-KO cell line used for genome-scale CRISPR-Cas9 screen for synthetic lethality. **b**. Representative live cell microscopy images of HeLa cells expressing GFP-PCNA used for quantifying duration of G2 phase. Loss of nuclear PCNA foci denotes entry into G2, while NEBD denotes exit from G2. Scale bar: 10 μm. **c**. Mitotic index of RPE1 wt and NIP45-KO cells following treatment with indicated doses of ICRF-193 (ICRF), SUMOi and/or ATMi+ATRi for 16 h in the presence of nocodazole (1 μM). Mitotic index was determined by immunostaining with phospho-MPM2 antibody and plotted as percent of nocodazole treatment alone (mean±s.d.; $n$ = 3 independent experiments; unpaired two-tailed $t$-test). **d**. Representative live cell microscopy images of HeLa wt and NIP45-KO cells expressing GFP-PCNA to monitor G2 length and mitotic progression following pre-treatment with SUMOi (150 nM) for 48 h and/or treatment with ICRF-193 (7 μM) immediately prior to imaging. DIC,

differential interference contrast. Scale bar: 10 μm. **e**. Quantification of live cell imaging tracking the mitotic fate of HeLa wt and NIP45-KO transiently expressing GFP-PCNA that were treated as in (d) (mean±s.d.; $n$ = 3 independent experiments; at least 30 cells were scored per condition per replicate). **f**. Immunoblot analysis of TOP2A levels in nuclear extracts from RPE1 wt and NIP45-KO cells left untreated or exposed to ICRF-193 (ICRF; 1 μM) for 4 h. **g**. Mitotic index of U2OS Flp-In T-REx wt, NIP45-KO and NIP45-KO/GFP-NIP45 cells following treatment with ICRF-193 (ICRF; 1 μM) and nocodazole (0.5 μM) for 16 h, determined as in (c) (mean±s.d.; $n$ = 3 independent experiments; unpaired two-tailed $t$-test comparing to NIP45-KO; U2OS WT: **$P$ = 0.0016; WT: *$P$ = 0.012; ΔSLD2: *$P$ = 0.043). **h**. Flow cytometry analysis of asynchronously growing HeLa wt and NIP45-KO cells exposed or not to IR (4 Gy) followed by 4 h incubation with nocodazole (150 ng/mL). Gated cells represent mitotic population. Data information: Data are representative of three (b) and two (a,f) independent experiments with similar outcome.

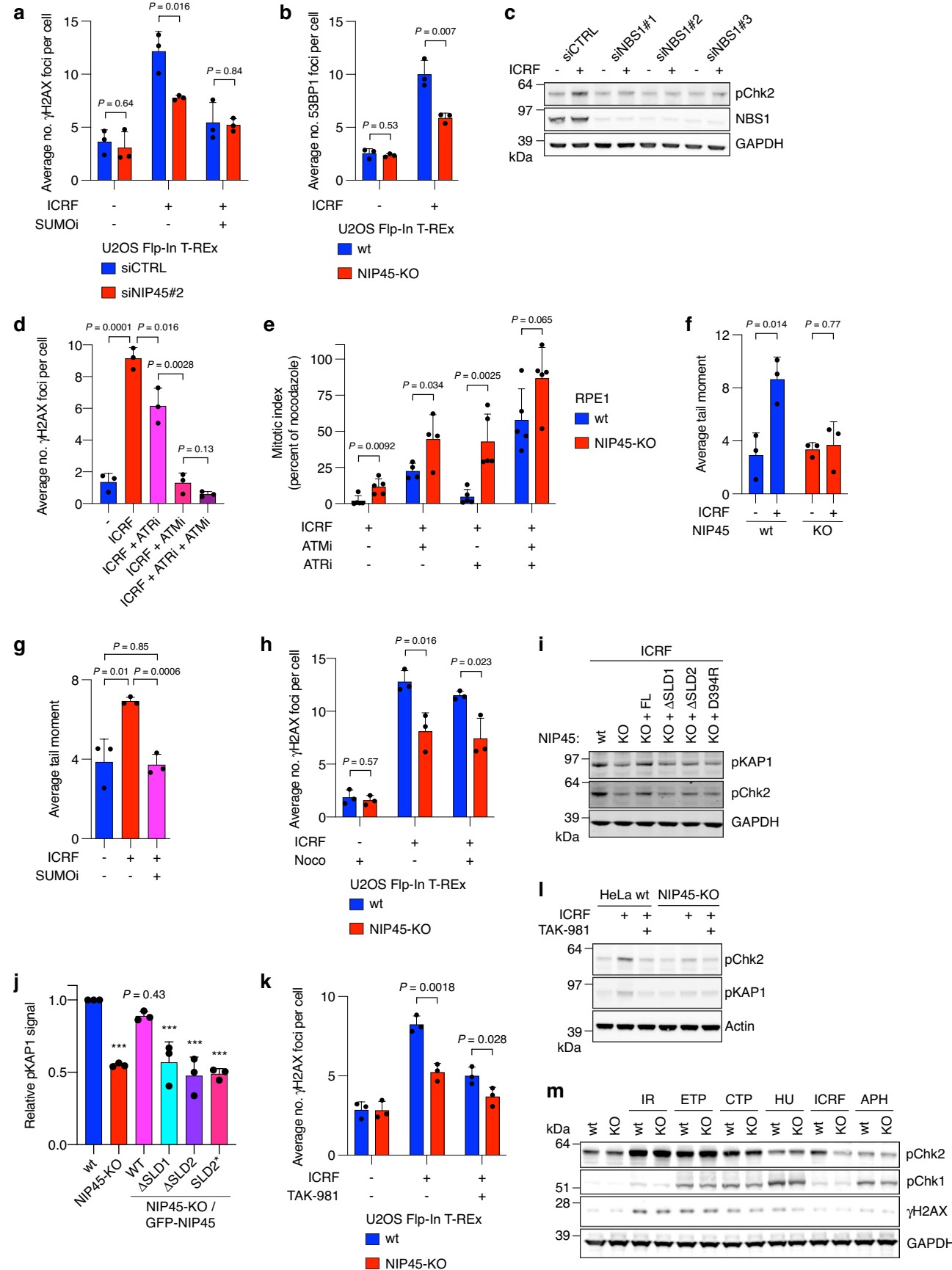

**Extended Data Fig. 5 | See next page for caption.**

**Extended Data Fig. 5 | Impact of NIP45 status and SUMOi on DSB formation upon ICRF-193 treatment. a**. Immunofluorescence analysis of γH2AX foci in U2OS Flp-In T-REx cells treated with ICRF-193 (ICRF; 1 μM) and/or SUMOi (2 μM) for 4 h following transfection with control (CTRL) or NIP45 siRNAs (mean±s.d.; $n$ = 3 independent experiments; unpaired two-tailed $t$-test). **b**. Immunofluorescence analysis of 53BP1 foci in U2OS Flp-In T-REx wt or NIP45-KO cells treated with ICRF-193 (ICRF; 1 μM) for 4 h (mean ± s.d.; $n$ = 3 independent experiments; unpaired two-tailed $t$-test). **c**. Immunoblot analysis of whole cell lysates from HeLa wt cells treated with ICRF-193 (7 μM) for 2 h following transfection with indicated siRNAs. **d**. As in (a), but cells were treated with ICRF-193 (1 μM) and/or ATRi and/or ATMi for 4 h (mean±s.d.; $n$ = 3 independent experiments; unpaired two-tailed $t$-test). **e**. Mitotic index, determined by immunostaining with phospho-MPM2 antibody and plotted as percentage of nocodazole treatment alone, of RPE1 wt and NIP45-KO cells treated with ICRF-193 (1 μM), and/or ATMi and/or ATRi for 16 h in presence of nocodazole (mean±s.d.; $n$ = 3 independent experiments; unpaired two-tailed $t$-test). **f**. DSB formation (tail moment) analyzed by comet assay in HeLa wt and NIP45-KO cells treated with ICRF-193 (25 μM) for 2 h (mean ± s.d.; $n$ = 3 independent experiments; unpaired two-tailed Mann-Whitney tests). **g**. As in (**f**), except that HeLa wt cells were treated with ICRF-193 (25 μM) and/or SUMOi (2 μM) for 2 h (mean±s.d.; $n$ = 3 independent experiments; unpaired two-tailed Mann-Whitney tests). **h**. As in (a), but cells were treated with ICRF-193 (1 μM) and/or nocodazole for 4 h (mean±s.d.; $n$ = 3 independent experiments; unpaired two-tailed $t$-test). **i**. Immunoblot analysis of whole cell lysates from U2OS Flp-In T-REx wt, NIP45-KO and NIP45-KO/GFP-NIP45 cells treated with ICRF-193 (2 μM) for 2 h. **j**. Quantification of pKAP1 signals in (i) (mean±s.d.; $n$ = 3 independent experiments; Tukey's multiple comparisons test; NIP45-KO: ***$P$ = 0.0001; GFP-NIP45 ΔSLD1: ***$P$ = 0.0002; GFP-NIP45 ΔSLD2: ***$P$ < 0.0001; GFP-NIP45 SLD2*: ***$P$ < 0.0001). **k**. As in (a), but cells were treated with ICRF-193 (1 μM) and/or TAK-981 (5 μM) for 4 h (mean±s.d.; $n$ = 3 independent experiments; unpaired two-tailed $t$-test). **l**. Immunoblot analysis of whole cell lysates from HeLa wt and NIP45-KO cells treated for 2 h with ICRF-193 (1 μM) and TAK-981 (5 μM). **m**. As in (l), but cells were treated with IR (4 Gy), etoposide (ETP), camptothecin (CTP), hydroxyurea (HU), ICRF-193 (1 μM), or aphidicolin (APH, 2 μM) for 2 h. Data information: Data are representative of three (i) and two (c,l,m) independent experiments with similar outcome.

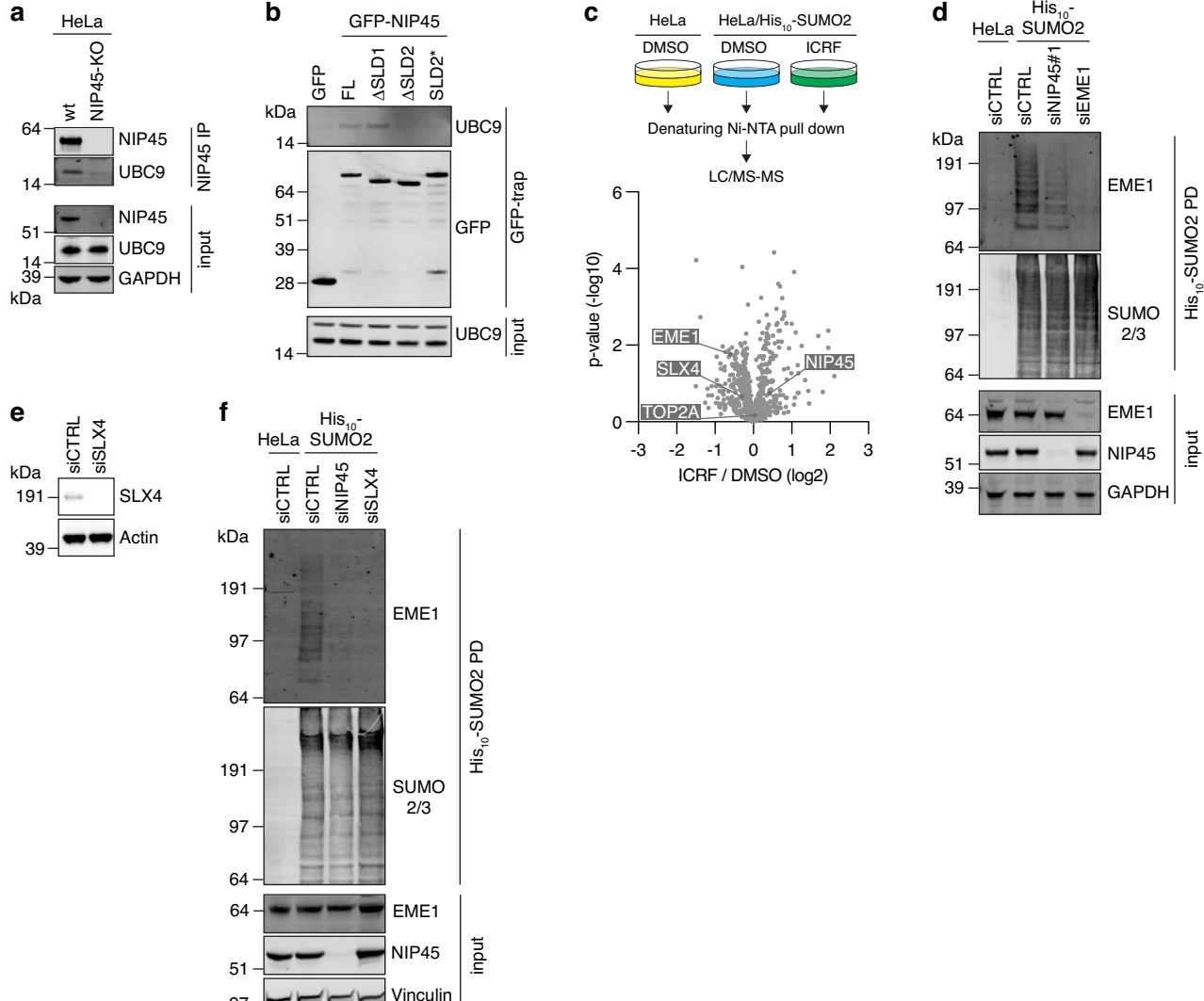

**Extended Data Fig. 6 | Impact of NIP45 and ICRF-193 on the SUMO system.**
**a**. Immunoblot analysis of NIP45 IPs from whole cell lysates of HeLa wt and
NIP45-KO cells. **b**. Immunoblot analysis of GFP IPs from whole cell lysates of
U2OS Flp-In T-REx NIP45-KO cells expressing GFP only or GFP-NIP45 alleles. **c**.
Mass spectrometry analysis of SUMOylated proteins isolated by denaturing His
(Ni-NTA) pulldown from HeLa or HeLa/His$_{10}$-SUMO2 cells treated with DMSO or
ICRF-193 (ICRF; 2 mM) for 2 h. Volcano plot show the mean difference of the protein
intensity, following subtraction of proteins identified in parental HeLa cells,
plotted against the *P* value (two-tailed two-sample Student's t-testing). Significant

differences (*q*-value < 0.05) were calculated by adjusting for multiple comparisons
with permutation-based FDR control (2,500 rounds of randomization) and
are indicated in blue (*n* = 4 biological replicates). **d**. Immunoblot analysis
of denaturing His (Ni-NTA) pulldown from HeLa or HeLa/His$_{10}$-SUMO2 cells
transfected with indicated siRNAs. **e**. Immunoblot analysis of whole cell lysates
from HeLa cells transfected with control (CTRL) or SLX4 siRNAs. **f**. Immunoblot
analysis of denaturing His (Ni-NTA) pulldown from HeLa or HeLa/His$_{10}$-SUMO2
cells transfected with indicated siRNAs. Data information: Data (a,b,d,e,f) are
representative of two independent experiments with similar outcome.

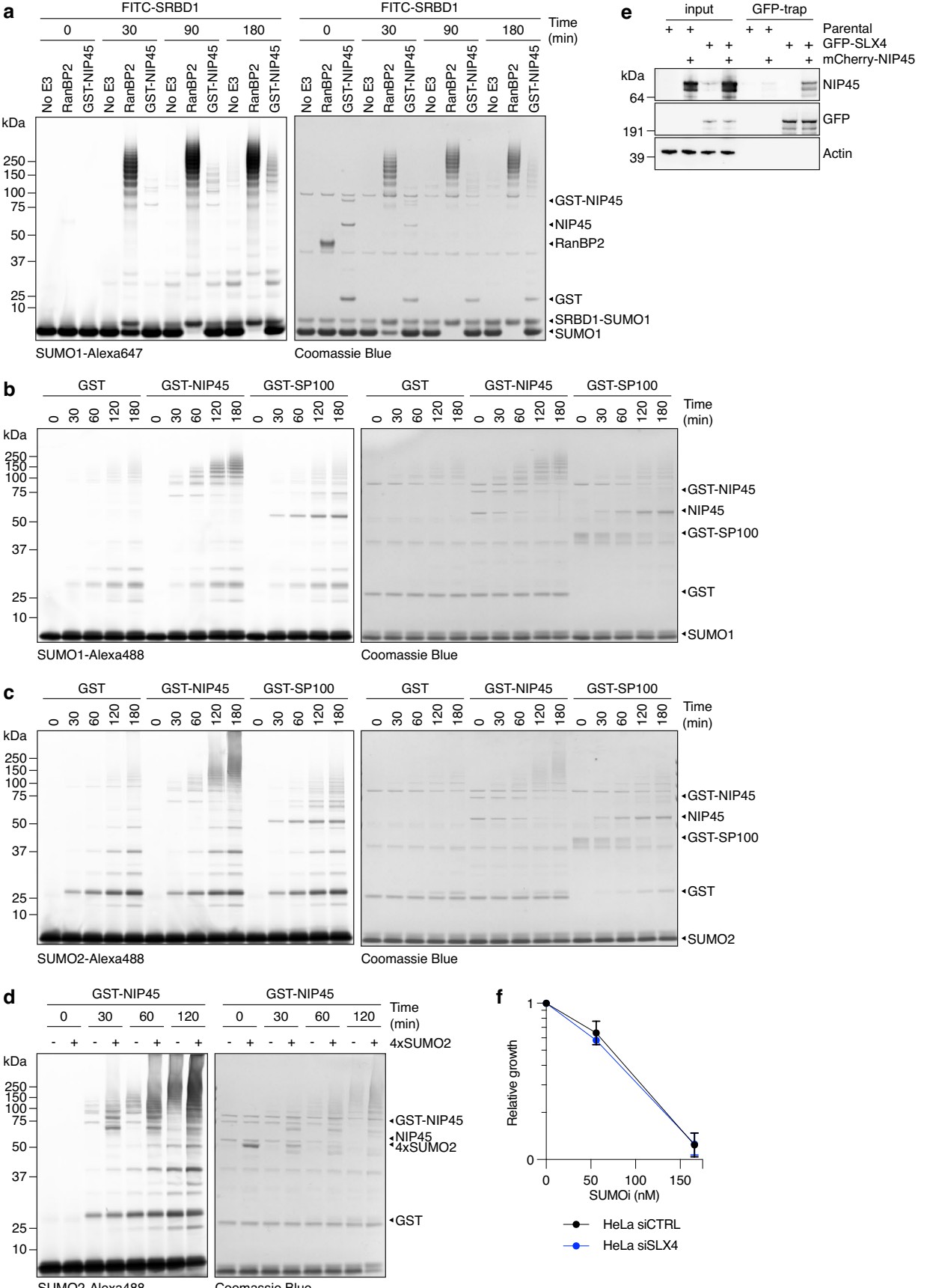

**Extended Data Fig. 7 | See next page for caption.**

**Extended Data Fig. 7 | NIP45 stimulates SUMO modification and interacts with SLX4. a**. *In vitro* SUMO modification assay containing purified SAE1/SAE2, UBC9, Alexa 647-labeled SUMO1 and FITC-labeled peptide containing a consensus SUMO modification site (FITC-SRBD1). Reactions without E3 ligase or with SUMO E3 ligase RanBP2(2532-2767) or GST-NIP45 were incubated at 37 °C for the indicated time. Reaction products were analyzed by SDS-PAGE and visualized by fluorescent scanning to detect Alexa 647 (left panel) or Coomassie blue staining (right panel). **b,c**. *In vitro* SUMO modification assays containing purified SAE1/SAE2, UBC9, Alexa 488-labeled SUMO1 (b) or SUMO2 (c) and either GST, GST-NIP45 or GST-SP100(241-360) were incubated at 37 °C for the indicated time.

Reaction products were analyzed by SDS-PAGE and visualized by fluorescent scanning to detect Alexa 488 (left panel) or Coomassie blue staining (right panel). **d**. As in (c), except that 4xSUMO2 was included in the reactions where indicated. **e**. Immunoblot analysis of GFP IPs from whole cell lysates of parental U2OS cells or U2OS cells expressing GFP-SLX4 transfected with plasmid encoding mCherry-NIP45. **f**. SRB cell growth assay using HeLa cells treated with indicated siRNAs and SUMOi doses (mean±s.d.; *n* = 3 independent experiments). Data information: Data are representative of three (e) and two (a-d) independent experiments with similar outcome.

# Reporting Summary

## Statistics

For all statistical analyses, confirm that the following items are present in the figure legend, table legend, main text, or Methods section.

| n/a | Confirmed | |
|---|---|---|
| ☐ | ☒ | The exact sample size (*n*) for each experimental group/condition, given as a discrete number and unit of measurement |
| ☐ | ☒ | A statement on whether measurements were taken from distinct samples or whether the same sample was measured repeatedly |
| ☐ | ☒ | The statistical test(s) used AND whether they are one- or two-sided *Only common tests should be described solely by name; describe more complex techniques in the Methods section.* |
| ☒ | ☐ | A description of all covariates tested |
| ☐ | ☒ | A description of any assumptions or corrections, such as tests of normality and adjustment for multiple comparisons |
| ☐ | ☒ | A full description of the statistical parameters including central tendency (e.g. means) or other basic estimates (e.g. regression coefficient) AND variation (e.g. standard deviation) or associated estimates of uncertainty (e.g. confidence intervals) |
| ☐ | ☒ | For null hypothesis testing, the test statistic (e.g. *F*, *t*, *r*) with confidence intervals, effect sizes, degrees of freedom and *P* value noted *Give P values as exact values whenever suitable.* |
| ☒ | ☐ | For Bayesian analysis, information on the choice of priors and Markov chain Monte Carlo settings |
| ☒ | ☐ | For hierarchical and complex designs, identification of the appropriate level for tests and full reporting of outcomes |
| ☒ | ☐ | Estimates of effect sizes (e.g. Cohen's *d*, Pearson's *r*), indicating how they were calculated |

*Our web collection on statistics for biologists contains articles on many of the points above.*

## Software and code

Policy information about availability of computer code

| Data collection | SoftWoRx (version 7.0.0; GE Healthcare), bcl2fastq (version 2.19.1), cutadapt (version 1.18), ImageStudio (version 3.1.4; LI-COR), ImageQuant LAS4000 software (version 1.2; GE Healthcare), Incucyte S3 Live-Cell Analysis System (version 2021A), BD FACS Diva software (version 9.0) |
|---|---|
| Data analysis | ScanR analysis software (version 2.8.1; Olympus), Spotfire (version 10.5.0; Tibco), SoftWoRx software (version 7.0.0; GE Healthcare), TriTek CometScore (version 2.0), GraphPad Prism (version 9.3.0), MaxQuant (version 1.5.3.30), Perseus (Tyanova et al., Nat Methods 13: 731-740 (2016)), R (https://www.r-project.org/), MAGeCK (version 0.5.8), drugZ (Colic et al., Genome Med 11: 52 (2019) ), BAGEL (Hart and Moffat, BMC Bioinformatics 17: 164 (2016) ), ImageStudio (version 3.1.4; LI-COR), ImageQuant LAS4000 software (version 1.2; GE Healthcare), FLUOstar Omega MARS (version V4.01; BMG Labtech), Incucyte S3 Live-Cell Analysis System (version 2021A), FCS Express (version 7; DeNovo Software), Fiji (version 2.3.0/1.53q) |

For manuscripts utilizing custom algorithms or software that are central to the research but not yet described in published literature, software must be made available to editors and reviewers. We strongly encourage code deposition in a community repository (e.g. GitHub). See the Nature Portfolio guidelines for submitting code & software for further information.

# Data

Policy information about availability of data

All manuscripts must include a data availability statement. This statement should provide the following information, where applicable:

- Accession codes, unique identifiers, or web links for publicly available datasets
- A description of any restrictions on data availability
- For clinical datasets or third party data, please ensure that the statement adheres to our policy

The CRISPR screen data sets (Supplementary Data 1-5) are provided with this study. The mass spectrometry proteomics data (Supplementary Data 6-7) have been deposited to the ProteomeXchange Consortium 67 via the Proteomics Identifications (PRIDE) partner repository (http://www.ebi.ac.uk/pride) under dataset ID PXD033739 (reviewer account: reviewer_pxd033739@ebi.ac.uk and password: MkZTXKw6). sgRNA sequences in the TKOv3 library are available on addgene (). All other data supporting the findings of this study are available within the article and supplementary information.

# Human research participants

Policy information about studies involving human research participants and Sex and Gender in Research.

| Reporting on sex and gender | N/A |
| --- | --- |
| Population characteristics | N/A |
| Recruitment | N/A |
| Ethics oversight | N/A |

Note that full information on the approval of the study protocol must also be provided in the manuscript.

# Field-specific reporting

Please select the one below that is the best fit for your research. If you are not sure, read the appropriate sections before making your selection.

☒ Life sciences ☐ Behavioural & social sciences ☐ Ecological, evolutionary & environmental sciences

For a reference copy of the document with all sections, see nature.com/documents/nr-reporting-summary-flat.pdf

# Life sciences study design

All studies must disclose on these points even when the disclosure is negative.

| Sample size | Data were obtained according to the field's best practice. No statistical method was used to predetermine sample size. For CRISPR screens, sample sizes were designed based on publications using the same cell lines and CRISPR libraries (Hart et al., G3 7: 2719-2727 (2017); Olivieri et al., Cell 182: 481-496.e21 (2020)). Sample size for each experiment is indicated either in figure legends or methods. |
| --- | --- |
| Data exclusions | No data were excluded from the analyses. |
| Replication | All experimental findings shown in this study were independently replicated at least twice with similar outcome. Information about replication is provided in the figure legends. |
| Randomization | The samples were not randomized. Randomization is generally not relevant for this study since we are working with cell populations and not test subjects. |
| Blinding | The investigators were not blinded to group allocation during data collection and analysis. For most experiments, data analysis was performed computationally with fixed parameters across samples. For practical reasons, investigators were not blinded when scoring number of UFBs per cell but great measure was taken to avoid bias. |

# Reporting for specific materials, systems and methods

We require information from authors about some types of materials, experimental systems and methods used in many studies. Here, indicate whether each material, system or method listed is relevant to your study. If you are not sure if a list item applies to your research, read the appropriate section before selecting a response.

## Materials & experimental systems

| n/a | Involved in the study |
|---|---|
| ☐ | ☒ Antibodies |
| ☐ | ☒ Eukaryotic cell lines |
| ☒ | ☐ Palaeontology and archaeology |
| ☒ | ☐ Animals and other organisms |
| ☒ | ☐ Clinical data |
| ☒ | ☐ Dual use research of concern |

## Methods

| n/a | Involved in the study |
|---|---|
| ☒ | ☐ ChIP-seq |
| ☐ | ☒ Flow cytometry |
| ☒ | ☐ MRI-based neuroimaging |

## Antibodies

| | |
|---|---|
| Antibodies used | The following commercially available antibodies were used: GAPDH (rabbit polyclonal, sc-25778, Santa Cruz; WB: 1:1,000), BLM (rabbit polyclonal, A300-110A, Bethyl Laboratories; WB: 1:1,000), Actin (mouse monoclonal clone C4, MAB1501, Merck; WB: 1:20,000), RMI2 (rabbit polyclonal, ab122685, Abcam; WB: 1:750), FKBP8 (rabbit monoclonal clone EPR7441(2), ab129113, Abcam; WB: 1:1,000), P300 (rabbit polyclonal, ab10485, Abcam; WB: 1:5,000), GFP (rabbit polyclonal, PABG1, Chromotek; WB: 1:2,000; IF: 1:2,000; mouse monoclonal clone 13.1, 11814460001, Merck; WB: 1:1,000), SUMO2/3 (rabbit polyclonal, ab3742, Abcam; WB: 1:1,000), Histone H3 (rabbit polyclonal, ab1791, Abcam; WB: 1:50,000), MUS81 (mouse monoclonal clone MTA30 2G10/3, sc-53382, Santa Cruz; WB: 1:250), ATM phospho-S1981 (rabbit monoclonal clone EP1890Y, ab81292, Abcam; WB: 1:1,000), ATM (mouse monoclonal clone 1B10, sc-135663, Santa Cruz; WB: 1:100), KAP1 phospho-S824 (rabbit polyclonal, A300-767A, Bethyl Laboratories; WB: 1:1,000), KAP1 (rabbit polyclonal, A300-274A, Bethyl Laboratories; WB: 1:1,000), Chk2 phospho-T68 (rabbit polyclonal, 2661, Cell Signaling Technologies; WB: 1:500), Chk2 (mouse monoclonal clone 8F12, MA5-31595, Invitrogen; WB: 1:500; rabbit monoclonal clone EPR4325, ab109413, Abcam, WB: 1:1,000), H2AX phospho-S139 (rabbit polyclonal, 2577, Cell Signaling Technologies; WB: 1:500; mouse monoclonal clone JBW301, 05-636, Merck, IF: 1:500), Chk1 phospho-S345 (rabbit polyclonal, 2348, Cell Signaling Technologies; WB: 1:1,000), Vinculin (mouse monoclonal clone hVIN-1, V9131, Merck; WB: 1:10,000), EME1 (mouse monoclonal clone MTA31 7h2/1, sc-53275, Santa Cruz; WB: 1:100), TOP2A (mouse monoclonal clone G-6, sc-166934, Santa Cruz; WB: 1:500), UBC9 (goat polyclonal, ab21193, Abcam; WB: 1:500), 53BP1 (rabbit polyclonal, NB100-304, Novus Biologicals; IF: 1:500), NBS1 (rabbit polyclonal, sc-11431, Santa Cruz; WB: 1:1000), SLX4 (rabbit polyclonal, ab100997, Abcam, WB: 1:1,000), phospho-Ser/Thr-Pro MPM-2 Cy5 conjugate (mouse monoclonal clone MPM-2, 16-220, Merck, IF: 1:500), Goat anti-Guinea Pig IgG Alexa Fluor 488 (A-11073, Invitrogen). The following custom-made antibodies were used: NIP45 (sheep polyclonal, raised against full-length human NIP45; WB: 1:1,000; IF: 1:1,000), RMI1 (mouse monoclonal clone TRR-56-8-3; WB: 1:1,000), PICH (guineapig polyclonal; WB: 1:200; IF: 1:500), SLX4 (rabbit polyclonal, gift from John Rouse, University of Dundee; WB: 1:1,000). |
| Validation | The specificity of antibodies against BLM, EME1, FKBP8, GFP, MUS81, NBS1, NIP45, P300, PICH, RMI1, RMI2, SUMO2/3 and SLX4 were validated using appropriate knockdown/knockout controls in this study (as shown in the manuscript). Other antibodies were used based on previous validation in the literature and/or manufacturer websites: Actin (Lessard, Cell Motil Cytoskeleton 10: 349-362 (1988)); Histone H3 (Abcam); ATM pS1981 (Abcam); ATM (Menotta et al., J Biol Chem 287: 41352–41363 (2012)); KAP1 pS824 (Bethyl Laboratories); KAP1 (Bethyl Laboratories); Chk2 pT68 (Cell Signaling Technologies); rabbit monoclonal Chk2 (Abcam); rabbit polyclonal H2AX pS193 (Cell Signaling Technologies); mouse monoclonal H2AX pS193 (Merck); Chk1 pS345 (Cell Signaling Technologies); Vinculin (Merck); TOP2A (Gothe et al., Mol Cell 75: 267-283.e12 (2019)); 53BP1 (Zelensky et al., PLoS Genet 16: e1008550 (2020)); pS/T MPM-2 (Davis et al., PNAS 80: 2926-2930 (1983)). The following antibodies were not validated to our knowledge: GAPDH, mouse monoclonal Chk2, UBC9. |

## Eukaryotic cell lines

Policy information about cell lines and Sex and Gender in Research

| | |
|---|---|
| Cell line source(s) | HeLa cells were obtained from ATCC. RPE1-hTERT PuroS cells were a kind gift from Andrew J. Holland. RPE1-hTERT BLM-KO and parental control cell line were kind gifts from Andrew Blackford. RPE1-hTERT p53-KO FLAG-Cas9 cells were a kid gift from Daniel Durocher. U2OS Flp-In T-REx cells were a kind gift from Helen Piwnica-Worms. U2OS Flp-In T-REx cells inducibly expressing GFP-SLX4 and parental control cells were a kind gift from John Rouse. Generation and validation of HeLa, RPE1-hTERT PuroS, RPE-hTERT p53-KO FLAG-Cas9 and U2OS Flp-In T-REx cell lines with targeted NIP45 KO or RIM1 KO are described in this study. |
| Authentication | Cell lines were not authenticated. |
| Mycoplasma contamination | All cell lines used in this study were regularly tested negative for mycoplasma infection. |
| Commonly misidentified lines (See ICLAC register) | Cell lines used in this study are not included in the ICLAC register of commonly misidentified cell lines. |

# Flow Cytometry

## Plots

Confirm that:

☒ The axis labels state the marker and fluorochrome used (e.g. CD4-FITC).

☒ The axis scales are clearly visible. Include numbers along axes only for bottom left plot of group (a 'group' is an analysis of identical markers).

☒ All plots are contour plots with outliers or pseudocolor plots.

☒ A numerical value for number of cells or percentage (with statistics) is provided.

## Methodology

| | |
|---|---|
| Sample preparation | Asynchronously growing HeLa wt and NIP45-KO cells were treated or not with IR (4 Gy) followed by nocodazole (150 ng/mL) for 4 h. Cells were collected and fixed in 70% ethanol at 4 °C and stained with phospho-MPM2 antibody (1:1.000) for 2 h at RT. |
| Instrument | Data was acquired using a 5 laser Becton Dickinson LSR Fortessa (488 nm, 561 nm, 355 nm, 405 nm and 640 nm). |
| Software | BD FACS Diva software (version 9.0) on the LSR Fortessa was used for data acquisition, and post-acquisition analysis was performed using FCS Express (version 7; DeNovo Software). |
| Cell population abundance | The cells used in this experiment were only analyzed and not sorted. |
| Gating strategy | A general gate was created around a population with similar forward and side scatter characteristics. Doublets and aggregates were then excluded using the propidium iodide pulse width measurement. Positive and negative populations were determined using staining controls. |

☒ Tick this box to confirm that a figure exemplifying the gating strategy is provided in the Supplementary Information.

