## [Peer Review File · Nature Structural & Molecular Biology]

Peer Review Information

Manuscript Title: The SUMO-NIP45 pathway processes toxic DNA catenanes to prevent mitotic failure

Corresponding author name(s): Niels Mailand, Emil Hertz

Reviewer Comments & Decisions:

Decision Letter, initial version:
--

Message: 22th Sep 2022

Dear Niels,

Thank you again for submitting your manuscript "The SUMO-NIP45 pathway processes toxic DNA catenanes to prevent mitotic failure". I again apologize for the delay in responding. Nevertheless, we now finally have comments (below) from all 3 reviewers who evaluated your paper. In light of those reports, we remain interested in your study and would like to see your response to the comments of the referees, in the form of a revised manuscript.

You will see that the reviewers acknowledge the quality and interest of the work. However, they all have detailed requests for increasing the mechanistic depth of the work, and editorially we agree that this would considerably strengthen the conclusions. Please be sure to address/respond to all concerns of the referees in full in a point-by-point response and highlight all changes in the revised manuscript text file.

We appreciate the requested revisions might be extensive, and we thus expect to see your revised manuscript within around 6 months. If you cannot send it within this time, please let us know. We will be happy to consider your revision as long as nothing similar has been accepted for publication at NSMB or published elsewhere. Should your manuscript be substantially delayed without notifying us in advance and your article is eventually published, the received date would be that of the revised, not the original, version.

As you already know, we put great emphasis on ensuring that the methods and statistics reported in our papers are correct and accurate. As such, if there are any changes that

should be reported, please submit an updated version of the Reporting Summary along with your revision.

Reporting Summary:

When submitting the revised version of your manuscript, please pay close attention to our [Digital Image Integrity Guidelines](https://www.nature.com/nature-portfolio/editorial-policies/image-integrity).

We require deposition of coordinates (and, in the case of crystal structures, structure factors) into the Protein Data Bank with the designation of immediate release upon publication (HPUB). Electron microscopy-derived density maps and coordinate data must be deposited in EMDB and released upon publication. Deposition and immediate release of NMR chemical shift assignments are highly encouraged. Deposition of deep sequencing and microarray data is mandatory, and the datasets must be released prior to or upon publication. To avoid delays in publication, dataset accession numbers must be supplied with the final accepted manuscript and appropriate release dates must be indicated at the galley proof stage. Please find the complete NRG policies on data availability at <http://www.nature.com/authors/policies/availability.html>.

Nature Structural & Molecular Biology is committed to improving transparency in authorship. As part of our efforts in this direction, we are now requesting that all authors identified as 'corresponding author' on published papers create and link their Open Researcher and Contributor Identifier (ORCID) with their account on the Manuscript Tracking System (MTS), prior to acceptance. This applies to primary research papers only. ORCID helps the scientific community achieve unambiguous attribution of all scholarly contributions. You can create and link your ORCID from the home page of the MTS by clicking on 'Modify my Springer Nature account'. For more information please visit please

visit www.springernature.com/orcid.

[Redacted]

Kind regards,
Florian

Dr Florian Ullrich
Associate Editor, Nature
Consulting Editor, Nature Structural & Molecular Biology
ORCID 0000-0002-1153-2040

Referee expertise:

Referee #1: SUMO signaling, proteomics

Referee #2: SUMO signaling and genome stability

Referee #3: genome stability and SUMO signaling

Reviewers' Comments:

Reviewer #1:

Remarks to the Author:

The evolutionary conserved SUMO system controls a wide variety of cellular signaling pathways and is essential for viability of both lower and higher eukaryotes. SUMO modification appears to be particularly important for proper mitotic progression, but the underlying mechanism has remained largely enigmatic. The recent availability of highly specific SUMO inhibitors (SUMOi) enabled novel approaches to tackle this question. Here, the authors initially performed a CRISPR screen to define genetic vulnerabilities in the context of low-dose SUMOi. They found that SUMOi sensitizes both HeLa and RPE-1 cells to inactivation of the BTRR complex, which is required for the disentanglement of catenated DNA and the resolution of anaphase DNA bridges in conjunction with the DNA helicase PICH. Subsequently, they revealed that loss of NIP45 generates a strong hypersensitivity to SUMOi and demonstrated that NIP1 deficiency is synthetic lethal with BTRR-PICH. Further work unravelled a pathway, where NIP45 acts as SUMOylation cofactor on selected targets by binding to Ubc9. Among the NIP45-induced SUMO targets is the nuclease SLX4. The authors propose that SUMOylation of SLX4 facilitates processing

of DNA catenanes thereby generating DSB and trigger a G2 arrest. Block of this pathway would lead to unresolved catenanes followed by defects in chromosome segregation and cytokinesis.

The proposed model is largely based on genetic data and still lacks some mechanistic insight. For example, it remains unclear how SUMOylation of SLX4 affects the processing of DNA catenanes. Further, it remains somewhat obscure, why NIP45 is a selective SUMOylation cofactor. Regardless of these issues, the experimental data are of very high quality and the work is of considerable novelty. The findings are of broad significance for basic understanding of SUMO biology, but have also a potential impact on cancer therapy. The manuscript is well written and the data presented support the hypothesis. I do have a few comments and suggestions that may help to further strengthen the work.

1. To further support the proposed model it would be important to determine how SLX4 deficiency affects cell viability upon SUMOi treatment? Further, it would be important to monitor cell viability upon co-treatment of ICRF and SUMOi. Along this line, it should also be tested how ICRF treatment affects cell viability in the NIP45 KO background.
2. Throughout the manuscript ML-792 was used as a SUMOi. ML-792 represents a first-generation SUMOi, a more advanced compound is TAK-981. Since TAK-981 is now already in Phase II clinical trials to treat solid and hematological trials, it would be interesting to see whether some key data can be recapitulated with TAK-981. Further, it has been widely demonstrated that SUMOi preferentially affects cell viability of cell lines with high levels of Myc. Can this be related to the current findings?
3. Is the SLD1 deletion mutant of NIP45 capable of binding Ubc9? As most of the rescue experiments (except Figure 2D) were shown with the NIP45 point mutant where the mutation is in the SLD2 domain, it would be interesting to know whether SLD1 deletion mutant loses the capacity to bind to Ubc9.
4. Figures 3C, 3D, and 5G. How do the authors explain the fact that NIP45 KO cells show appreciable UFBs but have negligible percentage of binucleation instead despite drastically reduced SUMOylation of SLX4?
5. How do the authors envision selective SUMOylation of NIP45 on only a handful of critical substrates? Does NIP45-Ubc9 also interact with the SLX4-MUS81-EME1 complex? Does SUMOylation of NIP45 itself have any impact on the SUMO conjugation of SLX4 and EME1?

Reviewer #2:

Remarks to the Author:

The manuscript from Hertz et al describes the combination of sumo E1 inhibitor and a series of genetic screens to identify the combinatorial role of sumo and NIP45 in directing breakage of catenated DNA products in G2. The work reveals that reduced sumo and/or loss of NIP45 are in turn lethal in the absence of the BTR-PITCH complex, which defends ultrafine bridges generated by incomplete decatenation. SUMOylation of SLX4/EME1 is NIP45 dependent and SLX4 is in part required for the DSB formation induced by ICRF.

The findings define the major players of the previously opaque TOP2-dependent 'decatenation checkpoint' and the work is thus a major contribution. The findings are also

important for those investigating inhibitors of the sumo pathway as anti-cancer agents.

The quality of the work presented is extremely high. Moreover, the amount of work here is considerable, combining five screens and two proteomic datasets (from five cell lines).

One area I feel needs a little clarification regards the role of NIP45. While I appreciate that an in-depth biochemical examination is beyond the scope of this already excellent paper, it is fascinating to consider, as shown, that NIP45 potentiates only a subset of UBC9 behaviours.

Is its interaction with UBC9 cell cycle-dependent? Is it acting like a cofactor to an E3 (no E3 is identified)? Are the areas of NIP45 unrelated to its conserved orthologues relevant to this pathway (perhaps pointing to a 'linking' role)? Does it interact with SLX4/EME1? Given the scope of the manuscript, I believe it would be churlish to want all of these questions answered – but two might be a reasonable expectation.

A curiosity-driven question: What's the explanation for the split in the cell populations of G2 length(fig 4E-G)? (It seems they are either long or short, but not in between).

The model in Fig5k doesn't bring anything - can the authors apply what they believe the data shows to the steps in the pathway and how NIP45 and SUMO contribute? (All models are to a degree wrong - and that's fine)

Reviewer #3:

Remarks to the Author:

The authors performed unbiased genome-scale CRISPR-Cas9 dropout screens in attempt to identify human genes that become essential upon mild inhibition of SUMOylation using ML-792, an inhibitor of SUMO-activating enzyme (SAE1-SAE2). Screens in HeLa and RPE-1 cell lines identified 74 and 53 genes, whose KO hypersensitizes cells to ML-792. Only 8 screen hits were shared among the two cell lines, including SAE1 (the non-catalytic subunit of the SUMO E1 enzyme heterodimer) and components of the BLM-TOP3A-RMI1-RMI2 (BTRR) complex, RMI1 and RMI2. The authors then focused on BTRR complex and the DNA helicase PICH that cooperate to resolve ultra-fine anaphase DNA bridges (UFBs), which form when interlinked DNA structure persist into mitosis. Authors showed that inhibiting SUMOylation leads to lethality of cells defective in BTRR-PICH pathway of handling UFBs. Then, authors switched attention to the top hit NFATC2IP (encoding the protein NIP45), whose KO conferred the strongest sensitivity to SUMOi in both screens. NIP45 is a highly conserved protein (Esc2 in *S. cerevisiae* and Rad60 in *S. pombe*) containing tandem C-terminal SUMO-like domains (SLDs) and implicated in genome stability maintenance through its ties to the SUMO pathway. Authors showed that both SLDs of NIP45 are essential for cell viability upon SUMOylation inhibition. To understand cellular function of NIP45 and its importance following SUMOylation impairment, authors subsequently performed synthetic lethality screens with NIP45 KO and discovered RMI1, RMI2, and BLM as shared hits between HeLa and RPE-1 screens. Moreover, NIP45 KO was synthetic lethal also with PICH KO in HeLa. Thus, SUMOylation, NIP45, involving its SLDs, and BTRR-PICH axis cooperate to support cell viability. Given the established role of BTRR-PICH in resolving UFBs, the authors then demonstrated that SUMOylation and NIP45 protect cells against UFB formation and binucleation. Treatment of cells with TOP2 catalytic inhibitor ICRF-193, but not aphidicolin, led to increased UFB levels in NIP45 KO, suggesting that it is particularly important for preventing UFBs arising from unresolved

DNA catenanes. Importantly, inhibition of SUMOylation with high dose of SUMOi alone led to phenotypes similar to those that result from inactivation of both NIP45 and BTRR-PICH, suggesting that SUMOylation is critical for suppressing UFBs and preventing binucleation.

Next, to address how NIP45 and SUMOylation counteract UFBs, authors conducted another CRISPR-Cas9 screen for genes synthetically lethal with RMI KO (component of BTRR-PICH). In addition to NIP45, the authors found the multi-nuclease scaffold protein SLX4 and its associated MUS81-EME1 nuclease complex components as hits in their screen, in addition to MYT1 kinase that together with WEE1 restricts mitotic entry via inhibitory phosphorylation of CDK1. This suggested that SUMOylation and NIP45 may counteract UFBs and binucleation by restraining mitotic entry, which was indeed confirmed by the authors using ICRF-193 mediated block of TOP2-dependent catenane resolution that restricts G2/M transition as readout. To address how SUMOylation and NIP45 promote G2 arrest when TOP2-dependent decatenation is inhibited by ICRF-193, authors focused on the markers of DNA double-strand break (DSB) induction reported previously to be induced upon ICRF-193 treatment. They could demonstrate that indeed inhibition of SUMOylation quantitatively suppressed ICRF-193-dependent DSB generation and signaling, paralleling their observations on ICRF-193-induced G2 arrest.

Finally, to identify substrates of SUMOylation mediated by NIP45 relevant for DSB induction upon ICRF-193 treatment, the authors performed mass spectrometry analysis of SUMO substrates comparing their abundance in NIP45 knockdown and siCTRL HeLa cells. Among those SUMOylated proteins that decreased in abundance in NIP45 knockdown cells the authors identified SLX4, EME1 and NIP45 itself, suggesting that it is also a SUMO substrate. The authors further confirmed that SUMOylation of both SLX4 and EME1 is strongly reduced in NIP45 depleted cells. Next, the authors assessed whether depletion of SLX4 and EME1 affects ICRF-193-dependent DSB generation. Loss of SLX4 but not of EME1 impaired ICRF-193-induced DSB formation. Collectively, these data suggested to the authors "that NIP45-dependent SUMOylation of SLX4 multi-nuclease complex, and perhaps additional factors, facilitates nucleolytic resolution of catenated DNA structures prior to mitotic entry to mitigate formation of UFBs and the threat they pose to chromosome segregation fidelity and cell fitness. This may contribute to the synthetic lethal interaction between RMI1 and SLX4".

Overall, the study is solid, well conducted and is based on the unbiased CRISPR-Cas9 and mass spectrometry screens. The authors clearly demonstrate that SUMOylation and NIP45 are collaborating with BTRR-PICH axis to prevent UFB accumulation and support cell viability. These findings are novel, of interest and worthy of publication.

However, the conclusion that it is indeed NIP45-dependent SUMOylation of the SLX4 multi-nuclease complex that facilitates nucleolytic resolution of catenated DNA structures prior to mitosis leading to their conversion into DSBs needs to be strengthened. At the moment, this is based merely on the correlations between the hits from various screens and the fact that SLX4 knockdown results in the similar phenotypes as loss of SUMOylation or NIP45. The authors cite a study (Guervilly et al., Mol Cell, 2015), in which SLX4 is reported to be a SUMO substrate that binds SUMO-charged UBC9 SUMO E2 via its SUMO-interacting motifs (SIMs). Moreover, SLX4 SIMs mediate its own SUMOylation as well as SUMO modification of its binding partner XPF. SLX4 variant harboring mutated SIMs (SLX4-SIMmut) is defective in its SUMOylation as well as in SUMOylation of XPF. Perhaps, SIMs of SLX4 in addition to recruiting SUMO-charged UBC9 also recruit NIP45 via its SLDs, thus promoting SUMOylation of SLX4? Loss of NIP45 in turn results in the

reduction of SLX4 SUMOylation. The authors could use SLX4-SIMmut defective in SUMOylation to study whether indeed the loss of SLX4 SUMOylation impairs ICRF-193-induced DSB formation. Alternatively, they can fuse catalytically active domain of a SUMO deconjugating enzyme (Ulp domain, UD fusion) to prevent SUMOylation of the SLX4 complex, as was done successfully previously to prevent SUMOylation of cohesin (Almedawar et al., Curr Biol, 2012), and study how loss of SLX4 SUMOylation impacts on ICRF-193-induced DSB formation. Further, the authors could address if RMI1 KO and SLX4-SIMmut are synthetically lethal.

As a minor comment, among the papers citing roles of Esc2 and Rad60 in genome integrity and in relation to the STE-BTRR complex, the authors should cite along Ref 48 and Refs 20-25, also Sollier et al, MCB 2009, where it is shown that Esc2, via its SUMO-like domains, acts similarly but in parallel to STR and forms distinct protein complexes.

Author Rebuttal to Initial comments

Point-by-point reply to the referees' comments

We thank the reviewers for their constructive and thoughtful feedback on our study. In the revised version of our manuscript, we have included the results of a range of new experiments performed on the basis of the reviewers' comments. Collectively, we believe the new additions to the manuscript address most of the referees' concerns and strengthen the key conclusion of our study that SUMOylation exerts an essential role in cell proliferation by enabling the resolution of toxic DNA catenanes via non-epistatic NIP45- and BTRR-PICH-dependent pathways to avert mitotic failure. In the following, we provide a detailed point-by-point response to the referee reports. The reviewers' comments (reproduced in their entirety) are shown in bold and italicized text, while our responses are in plain text. With the new additions, we hope the reviewers find our paper improved and suitable for publication in Nature Structural and Molecular Biology.

Reviewer #1:

The evolutionary conserved SUMO system controls a wide variety of cellular signaling pathways and is essential for viability of both lower and higher eukaryotes. SUMO modification appears to be particularly important for proper mitotic progression, but the underlying mechanism has remained largely enigmatic. The recent availability of highly specific SUMO inhibitors (SUMOi) enabled novel approaches to tackle this question. Here, the authors initially performed a CRISPR screen to define genetic vulnerabilities in the context of low-dose SUMOi. They found that SUMOi sensitizes both HeLa and RPE-1 cells to inactivation of the BTRR complex, which is required for the disentanglement of catenated DNA and the resolution of anaphase DNA bridges in conjunction with the DNA helicase PICH. Subsequently, they revealed that loss of NIP45 generates a strong hypersensitivity to SUMOi and demonstrated that NIP1 deficiency is synthetic lethal with BTRR-PICH. Further work unravelled a pathway, where NIP45 acts as SUMOylation cofactor on selected targets by binding to Ubc9. Among the NIP45-induced SUMO targets is the nuclease SLX4. The authors propose that SUMOylation of SLX4 facilitates processing of DNA catenanes thereby generating DSB and trigger a G2 arrest. Block of this pathway would lead to unresolved catenanes followed by defects in chromosome segregation and cytokinesis.

The proposed model is largely based on genetic data and still lacks some mechanistic insight. For example, it remains unclear how SUMOylation of SLX4 affects the processing of DNA catenanes. Further, it remains somewhat obscure, why NIP45 is a selective SUMOylation cofactor. Regardless of these issues, the experimental data are of very high quality and the work is of considerable novelty. The findings are of broad significance for basic understanding of SUMO biology, but have also a potential impact on cancer therapy. The manuscript is well written and the data presented support the hypothesis. I do have a few comments and suggestions that may help to further strengthen the work.

1. To further support the proposed model it would be important to determine how SLX4 deficiency affects cell viability upon SUMOi treatment? Further, it would be important to monitor cell viability upon co-treatment of ICRF and SUMOi. Along this line, it should also be tested how ICRF treatment affects cell viability in the NIP45 KO background.

We have added new data showing that cells grown in the presence of a low dose of SUMOi display increased sensitivity to ICRF-193 (new Figure S3E). By contrast, we found that NIP45-KO cells are not hypersensitive to ICRF-193 (Figure S2E). These findings agree well with our model (Figure 5K), as the BTRR-PICH pathway remains operational in NIP45-KO cells, but not in SUMOi-treated cells (where both the BTRR-PICH- and NIP45-driven catenane resolution mechanisms are impaired). Accordingly, although NIP45-KO cells have a defective ICRF-193-induced G2 checkpoint and thus enter mitosis with an increased level of unresolved catenanes manifesting as UFBs (Figure 3A,B; Figure 4E,F), the mitotic BTRR-PICH-mediated UFB resolution pathway ensures that these catenanes are processed prior to cell division, thereby preventing binucleation and an accompanying loss of proliferative potential (see also our response to point 4 below).

We also analyzed the impact of SLX4 knockdown on SUMOi sensitivity. Consistent with our SUMOi CRISPR screens, in which *SLX4* was not among gene knockouts displaying synthetic lethality with SUMOi (Figure 1A-C; Table S1; Table S2), we observed no increased sensitivity of SLX4-depleted cells to SUMOi (new Figure S7F). Together with the notion that SLX4 knockdown reduces ICRF-193-induced DSB formation, but not to the same extent as NIP45-KO (Figure 5J), this suggests that effectors other than SLX4-nuclease complexes can also contribute to catenane cleavage into DSBs downstream of NIP45. We now emphasize and discuss this conclusion in the revised manuscript (page 16-18).

2. Throughout the manuscript ML-792 was used as a SUMOi. ML-792 represents a first-generation SUMOi, a more advanced compound is TAK-981. Since TAK-981 is now already in Phase II clinical trials to treat solid and hematological trials, it would be interesting to see whether some key data can be recapitulated with TAK-981. Further, it has been widely demonstrated that SUMOi preferentially affects cell viability of cell lines with high levels of Myc. Can this be related to the current findings?

We agree that it is important to test whether key phenotypes observed with ML-792 are recapitulated by the more advanced and clinically relevant SUMO inhibitor TAK-981. In the revised manuscript, we include new data showing that NIP45-KO cells are exquisitely sensitive to TAK-981 (new Figure S2B), mirroring the effect of ML-792. We also show that TAK-981 treatment phenocopies the impact of ML-792 in suppressing ICRF193-induced DSB formation, as evidenced by γ H2AX foci formation and phosphorylation of KAP1 and CHK2 (new Figure S5K,L).

To examine the potential relevance of c-Myc expression on cellular SUMOi hypersensitivity and the synthetic lethality relationship between NIP45 and SUMOi that we observed in several cell lines (RPE1, U2OS and HeLa; Figure 1B,C,J; Figure 2D; Figure S2A), we probed c-Myc abundance in these cell lines. As shown in Figure R1 below, we found that c-Myc levels differed considerably between RPE1, U2OS and HeLa cells, being very low in non-transformed RPE1 cells and high in HeLa cells. However, deletion of NIP45 did not detectably impact c-Myc expression in any of these cell lines. These observations suggest there is no obvious correlation between c-Myc expression levels and the synthetic lethality relationship between NIP45 and SUMOi.

Figure R1.

c-Myc expression levels in different cell lines and impact of NIP45 deletion

Whole cell lysates of the indicated cell lines were immunoblotted with antibodies to c-Myc (9E10) and Vinculin (loading control).

3. Is the SLD1 deletion mutant of NIP45 capable of binding Ubc9? As most of the rescue experiments (except Figure 2D) were shown with the NIP45 point mutant where the mutation is in the SLD2 domain, it would be interesting to know whether SLD1 deletion mutant loses the capacity to bind to Ubc9.

We show in Figure S6B that unlike the NIP45 SLD2 mutants, the SLD1 deletion mutant is proficient for binding to UBC9. We do not yet know the precise function of the SLD1, which like SLD2 is essential for the role of NIP45 in protecting against SUMOi hypersensitivity and promoting ICRF193-induced DSB formation (Figure 2B-D; Figure S5I,J), but this is an interesting question for future studies.

4. Figures 3C, 3D, and 5G. How do the authors explain the fact that NIP45 KO cells show appreciable UFBs but have negligible percentage of binucleation instead despite drastically reduced SUMOylation of SLX4?

Our data suggest that cells lacking NIP45 become critically dependent on the BTRR-PICH-mediated mitotic UFB resolution pathway to avoid binucleation caused by unresolved catenanes. In the absence of the NIP45-dependent pathway for resolving catenanes, it is true that cells enter mitosis with a significantly increased level of catenated DNA structures manifesting as UFBs (Figure 3A,B). However, these catenanes can be resolved by BTRR-PICH in mitosis to prevent binucleation, providing a rationale for why NIP45-KO alone does not lead to binucleation despite giving rise to an increased level of UFBs.

5. How do the authors envision selective SUMOylation of NIP45 on only a handful of critical substrates? Does NIP45-Ubc9 also interact with the SLX4-MUS81-EME1 complex? Does SUMOylation of NIP45 itself has any impact of the SUMO conjugation of SLX4 and EME1?

How NIP45 promotes SUMOylation of a small number of specific target proteins in conjunction with UBC9 is an interesting question. While it is possible that NIP45 could act as a cofactor for a specific SUMO E3 ligase to promote SUMOylation of a subset of its substrates, neither our CRISPR screens nor our NIP45 interactome data revealed any such candidate SUMO E3. Accordingly, given that NIP45 interacts with UBC9 via the SLD2

(Figure 5E; Figure S6B), we considered the possibility that NIP45 may itself function as a SUMO E3 ligase. We observed that similar to the SUMO E3 ligase RanBP2, recombinant NIP45 underwent extensive auto-SUMOylation in the presence of UBC9 *in vitro*, in a reaction that was considerably more efficient than the modification of the optimal SUMO substrate SP100 under the same assay conditions (new Figure S7A-D). Interestingly, while NIP45 promoted polymerization of both free SUMO1 and SUMO2, it was much more efficient at modifying a linearly fused 4xSUMO2 protein mimicking a poly-SUMO2 chain (new Figure S7B-D). In fact, in the presence of 4xSUMO2, NIP45-mediated SUMO conjugation was shifted from auto-modification towards SUMOylation of this substrate (new Figure S7D). These findings are consistent with NIP45 acting as a specialized SUMO E3 ligase with SUMO chain extension activity. However, unlike RanBP2, NIP45 did not stimulate UBC9-mediated modification of peptides containing SUMO consensus sites (new Figure S7A). This suggests that NIP45 might have a narrow substrate preference, perhaps restricted to its native interaction partners, in keeping with the small range of proteins including SLX4 whose SUMOylation depends on NIP45 (Figure 5F). Consistent with this idea, we found that NIP45 and SLX4 interact in cells (new Figure 5H; new Figure S7E). While EME1 SUMOylation is also dependent on NIP45 (Figure S6D), we observed no detectable interaction of NIP45 with MUS81 and EME1 (data not shown), but it seems likely that the binding of NIP45 to SLX4 may bring NIP45-UBC9 into proximity of EME1 to facilitate its efficient SUMOylation, supported by the notion that EME1 SUMOylation is SLX4-dependent (Figure S6E,F; reported also by (Guervilly et al., Mol Cell (2015))).

Our data show that NIP45 is SUMOylated in cells (Figure 5F) and undergoes extensive auto-SUMOylation *in vitro* (new Figure S7A-D). Hence, it is formally possible that NIP45 SUMOylation status could impact its function in promoting SLX4 and EME1 SUMOylation, although we note that SLX4 interacts with non-SUMOylated NIP45 in cells (new Figure S7E). However, despite many attempts, we have not been able to generate a SUMOylation-deficient NIP45 mutant that would enable us to test this possibility directly, likely due to NIP45 auto-SUMOylation being targeted promiscuously to many lysine residues. Accordingly, addressing this point would be a major undertaking that is not feasible within the time frame of our revisions. However, we believe the main conclusions of our study are not compromised by the lack of such data.

Reviewer #2:

The manuscript from Hertz et al describes the combination of sumo E1 inhibitor and a series of genetic screens to identify the combinatorial role of sumo and NIP45 in directing breakage of catenated DNA products in G2. The work reveals that reduced sumo and or loss of NIP45 are in turn lethal in the absence of the BTR-PITCH complex, which defends ultrafine bridges generated by incomplete decatenation. SUMOylation of SLX4/EME1 is NIP45 dependent and SLX4 is in part required for the DSB formation induced by ICRF.

The findings define the major players of the previously opaque TOP2-dependent 'decatenation checkpoint' and the work is thus a major contribution. The findings are also important for those investigating inhibitors of the sumo pathway as anti-cancer agents.

The quality of the work presented is extremely high. Moreover, the amount of work here is considerable, combining five screens and two proteomic datasets (from five cell lines).

One area I feel needs a little clarification regards the role of NIP45. While I appreciate that an in-depth biochemical examination is beyond the scope of this already excellent paper, it is fascinating to consider, as shown, that NIP45 potentiates only a subset of UBC9 behaviours. Is its interaction with UBC9 cell cycle-dependent? Is it acting like a cofactor to an E3 (no E3 is identified)? Are the areas of NIP45 unrelated to its conserved orthologues relevant to this pathway (perhaps pointing to a ‘linking’ role)? Does it interact with SLX4/EME1? Given the scope of the manuscript, I believe it would be churlish to want all of these questions answered – but two might be a reasonable expectation.

How NIP45 promotes SUMOylation of a small number of specific target proteins in conjunction with UBC9 is an interesting question. While it is possible that NIP45 could act as a cofactor for a specific SUMO E3 ligase to promote SUMOylation of a subset of its substrates, neither our CRISPR screens nor our NIP45 interactome data revealed any such candidate SUMO E3. Accordingly, given that NIP45 interacts with UBC9 via the SLD2 (Figure 5E; Figure S6B), we considered the possibility that NIP45 may itself function as a SUMO E3 ligase. We observed that similar to the SUMO E3 ligase RanBP2, recombinant NIP45 underwent extensive auto-SUMOylation in the presence of UBC9 *in vitro*, in a reaction that was considerably more efficient than the modification of the optimal SUMO substrate SP100 under the same assay conditions (new Figure S7A-D). Interestingly, while NIP45 promoted polymerization of both free SUMO1 and SUMO2, it was much more efficient at modifying a linearly fused 4xSUMO2 protein mimicking a poly-SUMO2 chain (new Figure S7B-D). In fact, in the presence of 4xSUMO2, NIP45-mediated SUMO conjugation was shifted from auto-modification towards SUMOylation of this substrate (new Figure S7D). These findings are consistent with NIP45 acting as a specialized SUMO E3 ligase with SUMO chain extension activity. However, unlike RanBP2, NIP45 did not stimulate UBC9-mediated modification of peptides containing SUMO consensus sites (new Figure S7A). This suggests that NIP45 might have a narrow substrate preference, perhaps restricted to its native interaction partners, in keeping with the small range of proteins including SLX4 whose SUMOylation depends on NIP45 (Figure 5F). Consistent with this idea, we found that NIP45 and SLX4 interact in cells (new Figure 5H; new Figure S7E). While EME1 SUMOylation is also dependent on NIP45 (Figure S6D), we observed no detectable interaction of NIP45 with MUS81 and EME1 (data not shown), but it seems likely that the binding of NIP45 to SLX4 may bring NIP45-UBC9 into proximity of EME1 to facilitate its efficient SUMOylation, supported by the notion that EME1 SUMOylation is SLX4-dependent (Figure S6E,F; reported also by (Guervilly et al., Mol Cell (2015))).

We also investigated whether areas of NIP45 other than the SLDs are important for its role in promoting SUMO_i resistance and DSB formation upon ICRF-193 treatment. AlphaFold predicted that most of the N-terminal half of human NIP45 is disordered, except for an alpha helix predicted with high confidence (amino acids 209-233; <https://alphafold.ebi.ac.uk/entry/Q8NCF5>). We therefore stably reconstituted NIP45-KO cells either with exogenous NIP45 lacking the portion N-terminal to the predicted alpha helix (deletion of amino acids 1-207; ΔN1 mutant) or with NIP45 lacking the full region N-terminal to the SLDs (deletion of amino acids 1-260; ΔN2 mutant) (Figure 2B; new Figure S2H). We then assessed the ability of these mutants to reverse phenotypes resulting from loss of endogenous NIP45. We found that the NIP45 ΔN1 mutant, but not NIP45 ΔN2, rescued the SUMO_i hypersensitivity arising from loss of endogenous NIP45 (new Figure S2I). This suggests that, like the SLDs, the putative N-terminal alpha helical domain is critical for the function of NIP45 in protecting against SUMO_i cytotoxicity, possibly by mediating protein-

protein interactions. Future studies will be required to establish the precise function of this domain.

A curiosity-driven question: What's the explanation for the split in the cell populations of G2 length(fig 4E-G)? (It seems they are either long or short, but not in between).

We speculate that the observed split in G2 duration (long or short) in cells treated with ICRF-193 may reflect the fact that catenanes likely represent undamaged DNA structures that in themselves do not trigger ATM/ATR-dependent G2 arrest. However, they can elicit such a response when converted into DSBs by the action of the SUMO-NIP45 pathway. Thus, this checkpoint may operate in a switch-like fashion, being either off or on, with the latter state triggering a lengthy ATM/ATR-dependent block to mitotic entry. When SUMOi concentrations exceed a critical threshold required to suppress SUMO-dependent catenane conversion into DSBs via NIP45 (above 37.5 nM; Figure 4E), the ATM/ATR-dependent G2 checkpoint is no longer activated, hence ICRF-193-treated cells pass through G2 with kinetics paralleling that of untreated cells. At lower SUMOi concentrations close to this threshold (around 37.5 nM; Figure 4E) the G2 checkpoint may be suppressed in some cells but not others, giving rise to the bimodal distribution of G2 length observed under these conditions.

The model in Fig5k doesn't bring anything - can the authors apply what they believe the data shows to the steps in the pathway and how NIP45 and SUMO contribute? (All models are to a degree wrong - and that's fine)

We have redrawn the model (Figure 5K) in order to better illustrate how based on our data we believe SUMO, NIP45 and BTRR-PICH act in parallel catenane resolution pathways operating in interphase and mitosis to promote faithful cell division.

Reviewer #3:

The authors performed unbiased genome-scale CRISPR-Cas9 dropout screens in attempt to identify human genes that become essential upon mild inhibition of SUMOylation using ML-792, an inhibitor of SUMO-activating enzyme (SAE1-SAE2). Screens in HeLa and RPE-1 cell lines identified 74 and 53 genes, whose KO hypersensitizes cells to ML-792. Only 8 screen hits were shared among the two cell lines, including SAE1 (the non-catalytic subunit of the SUMO E1 enzyme heterodimer) and components of the BLM-TOP3A-RMI1-RMI2 (BTRR) complex, RMI1 and RMI2. The authors then focused on BTRR complex and the DNA helicase PICH that cooperate to resolve ultra-fine anaphase DNA bridges (UFBs), which form when interlinked DNA structure persist into mitosis. Authors showed that inhibiting SUMOylation leads to lethality of cells defective in BTRR-PICH pathway of handling UFBs. Then, authors switched attention to the top hit NFATC2IP (encoding the protein NIP45), whose KO conferred the strongest sensitivity to SUMOi in both screens. NIP45 is a highly conserved protein (Esc2 in *S. cerevisiae* and Rad60 in *S. pombe*) containing tandem C-terminal SUMO-like domains (SLDs) and implicated in genome stability maintenance through its ties to the SUMO pathway. Authors showed that both SLDs of NIP45 are essential for cell viability upon SUMOylation inhibition. To understand cellular function of NIP45 and its importance following SUMOylation impairment, authors subsequently performed synthetic lethality screens with

NIP45 KO and discovered RMI1, RMI2, and BLM as shared hits between HeLa and RPE-1 screens. Moreover, NIP45 KO was synthetic lethal also with PICH KO in HeLa. Thus, SUMOylation, NIP45, involving its SLDs, and BTRR-PICH axis cooperate to support cell viability. Given the established role of BTRR-PICH in resolving UFBs, the authors then demonstrated that SUMOylation and NIP45 protect cells against UFB formation and binucleation. Treatment of cells with TOP2 catalytic inhibitor ICRF-193, but not aphidicolin, led to increased UFB levels in NIP45 KO, suggesting that it is particularly important for preventing UFBs arising from unresolved DNA catenanes. Importantly, inhibition of SUMOylation with high dose of SUMOi alone led to phenotypes similar to those that result from inactivation of both NIP45 and BTRR-PICH, suggesting that SUMOylation is critical for suppressing UFBs and preventing binucleation.

Next, to address how NIP45 and SUMOylation counteract UFBs, authors conducted another CRISPR-Cas9 screen for genes synthetically lethal with RMI KO (component of BTRR-PICH). In addition to NIP45, the authors found the multi-nuclease scaffold protein SLX4 and its associated MUS81-EME1 nuclease complex components as hits in their screen, in addition to MYT1 kinase that together with WEE1 restricts mitotic entry via inhibitory phosphorylation of CDK1. This suggested that SUMOylation and NIP45 may counteract UFBs and binucleation by restraining mitotic entry, which was indeed confirmed by the authors using ICRF-193 mediated block of TOP2-dependent catenane resolution that restricts G2/M transition as readout. To address how SUMOylation and NIP45 promote G2 arrest when TOP2-dependent decatenation is inhibited by ICRF-193, authors focused on the markers of DNA double-strand break (DSB) induction reported previously to be induced upon ICRF-193 treatment. They could demonstrate that indeed inhibition of SUMOylation quantitatively suppressed ICRF-193-dependent DSB generation and signaling, paralleling their observations on ICRF-193-induced G2 arrest.

Finally, to identify substrates of SUMOylation mediated by NIP45 relevant for DSB induction upon ICRF-193 treatment, the authors performed mass spectrometry analysis of SUMO substrates comparing their abundance in NIP45 knockdown and siCTRL HeLa cells. Among those SUMOylated proteins that decreased in abundance in NIP45 knockdown cells the authors identified SLX4, EME1 and NIP45 itself, suggesting that it is also a SUMO substrate. The authors further confirmed that SUMOylation of both SLX4 and EME1 is strongly reduced in NIP45 depleted cells. Next, the authors assessed whether depletion of SLX4 and EME1 affects ICRF-193-dependent DSB generation. Loss of SLX4 but not of EME1 impaired ICRF-193-induced DSB formation. Collectively, these data suggested to the authors “that NIP45-dependent SUMOylation of SLX4 multi-nuclease complex, and perhaps additional factors, facilitates nucleolytic resolution of catenated DNA structures prior to mitotic entry to mitigate formation of UFBs and the threat they pose to chromosome segregation fidelity and cell fitness. This may contribute to the synthetic lethal interaction between RMI1 and SLX4”.

Overall, the study is solid, well conducted and is based on the unbiased CRISPR-Cas9 and mass spectrometry screens. The authors clearly demonstrate that SUMOylation and NIP45 are collaborating with BTRR-PICH axis to prevent UFB accumulation and support cell viability. These findings are novel, of interest and worthy of publication.

However, the conclusion that it is indeed NIP45-dependent SUMOylation of the SLX4 multi-nuclease complex that facilitates nucleolytic resolution of catenated DNA structures prior to mitosis leading to their conversion into DSBs needs to be strengthened. At the

moment, this is based merely on the correlations between the hits from various screens and the fact that SLX4 knockdown results in the similar phenotypes as loss of SUMOylation or NIP45. The authors cite a study (Guervilly et al., Mol Cell, 2015), in which SLX4 is reported to be a SUMO substrate that binds SUMO-charged UBC9 SUMO E2 via its SUMO-interacting motifs (SIMs). Moreover, SLX4 SIMs mediate its own SUMOylation as well as SUMO modification of its binding partner XPF. SLX4 variant harboring mutated SIMs (SLX4-SIMmut) is defective in its SUMOylation as well as in SUMOylation of XPF. Perhaps, SIMs of SLX4 in addition to recruiting SUMO-charged UBC9 also recruit NIP45 via its SLDs, thus promoting SUMOylation of SLX4? Loss of NIP45 in turn results in the reduction of SLX4 SUMOylation. The authors could use SLX4-SIMmut defective in SUMOylation to study whether indeed the loss of SLX4 SUMOylation impairs ICRF-193-induced DSB formation. Alternatively, they can fuse catalytically active domain of a SUMO deconjugating enzyme (Ulp domain, UD fusion) to prevent SUMOylation of the SLX4 complex, as was done successfully previously to prevent SUMOylation of cohesin (Almedawar et al., Curr Biol, 2012), and study how loss of SLX4 SUMOylation impacts on ICRF-193-induced DSB formation. Further, the authors could address if RMI1 KO and SLX4-SIMmut are synthetically lethal.

We thank the Reviewer for the constructive and insightful suggestions. We extended the insights into how NIP45 promotes SUMOylation of SLX4 by including new data showing that NIP45 and SLX4 interact in cells (new Figure 5H; new Figure S7E) and that NIP45 acts as a specialized SUMO E3 ligase with SUMO chain extension activity *in vitro* (new Figure S7A-D). Employing an SLX4 SIM mutant that has previously been shown to be deficient for SUMOylation (Guervilly et al., Mol Cell 2015) to test whether SLX4 SUMOylation is important for promoting NIP45-dependent catenane conversion into DSBs upon ICRF-193 treatment is an excellent idea. However, despite very extensive efforts, we have been unable to obtain conclusive results using this approach, as exogenously expressed WT and SIM-mutated forms of human SLX4 consistently did not behave well in our hands and were highly prone to aggregation in cells, possibly due to the very large size of SLX4, thus raising doubts about the functionality of these exogenous SLX4 proteins (our unpublished observations). Indeed, we found that SLX4 overexpression is associated with considerable cytotoxicity, as has been reported previously (Guervilly et al., Mol Cell 2015). This may be at least partially due to a high level of spontaneous DSB formation seen in cells expressing exogenous SLX4 proteins (our unpublished observations), which could obscure the potential impact of a SUMOylation-deficient SLX4 protein on ICRF-193-induced DSB formation. Consequently, our attempts at expressing and functionally characterizing a fusion protein between full-length SLX4 and the catalytic domain of the human SUMO-specific protease SENP6 were also unsuccessful. The inherent limitations imposed by these technical challenges associated with expression of exogenous human SLX4 proteins in cells thus unfortunately precluded us from testing directly whether SLX4 SUMOylation is important for ICRF-193-induced DSB formation via the NIP45 pathway. Our data clearly show that SLX4 SUMOylation is NIP45-dependent and that the ability of NIP45 to promote target SUMOylation via its binding to UBC9 by the SLD2 is instrumental for its function in ICRF-193-induced DSB formation and G2 arrest (Figure 4F; Figure 5E-G; Figure S5I,J). However, although SLX4 knockdown partially recapitulates the impact of NIP45-KO in suppressing ICRF-193-induced DSB formation (Figure 5J), we cannot formally demonstrate at this point that NIP45-dependent SLX4 SUMOylation, and not just SLX4 *per se*, is important for the conversion of catenanes into DSBs prior to mitosis. The notion that SLX4 knockdown reduces ICRF-193-induced DSB formation, but not to the full extent seen for NIP45-KO (Figure 5J), suggests that effectors other than SLX4-nuclease complexes may also contribute to the conversion of

catenanes into DSBs downstream of NIP45. We now emphasize and discuss these important points in the manuscript (page 16-18).

As a minor comment, among the papers citing roles of Esc2 and Rad60 in genome integrity and in relation to the STE-BTRR complex, the authors should cite along Ref 48 and Refs 20-25, also Sollier et al, MCB 2009, where it is shown that Esc2, via its SUMO-like domains, acts similarly but in parallel to STR and forms distinct protein complexes.

We apologize for the oversight and now cite this reference in the revised manuscript.

Decision Letter, first revision:

Message: Our ref: NSMB-A46539A

22nd Mar 2023

Dear Dr. Mailand,

Thank you for submitting your revised manuscript "The SUMO-NIP45 pathway processes toxic DNA catenanes to prevent mitotic failure" (NSMB-A46539A). It has now been seen by the original referees and their comments are below. The reviewers find that the paper has improved in revision, and therefore we'll be happy in principle to publish it in Nature Structural & Molecular Biology, pending minor revisions to comply with our editorial and formatting guidelines.

Sincerely,

Carolina

Carolina Perdigoto, PhD
Chief Editor
Nature Structural & Molecular Biology
orcid.org/0000-0002-5783-7106

Reviewer #1 (Remarks to the Author):

The authors have adequately addressed the points I had raised on their initial version. This work is now a very comprehensive and data-rich manuscript, which I can strongly recommend for publication without any hesitation.

Reviewer #2 (Remarks to the Author):

The authors have addressed the main concerns and built a considerable mechanistic story on how SUMOylation and NIP45 regulate DNA decatenation/DSB. The work is an impressive and important contribution.

Reviewer #3 (Remarks to the Author):

This is a novel and important manuscript, with very solid data and very high quality experiments. The authors further improved the manuscript during the revision, trying to accommodate diverse points. In my view, the manuscript is suitable for publication in its current form.

Dana Brnzei

Final Decision Letter:

Message 26th Jun 2023

:

Dear Dr. Mailand,

Please accept my sincere apologies for the delay in sending you the final decision on your study - I am afraid I have been on medical leave for the past weeks.

We are now happy to accept your revised paper "The SUMO-NIP45 pathway processes toxic DNA catenanes to prevent mitotic failure" for publication as a Article in Nature Structural & Molecular Biology.

Your paper will be published online soon after we receive proof corrections and will appear in print in the next available issue. You can find out your date of online publication by contacting the production team shortly after sending your proof corrections. Content is published online weekly on Mondays and Thursdays, and the embargo is set at 16:00 London time (GMT)/11:00 am US Eastern time (EST) on the day of publication. Now is the time to inform your Public Relations or Press Office about your paper, as they might be interested in promoting its publication. This will allow them time to prepare an accurate and satisfactory press release. Include your manuscript tracking number (NSMB-A46539B) and our journal name, which they will need when they contact our press office.

About one week before your paper is published online, we shall be distributing a press release to news organizations worldwide, which may very well include details of your work. We are happy for your institution or funding agency to prepare its own press release, but it must mention the embargo date and Nature Structural & Molecular Biology. If you or your Press Office have any enquiries in the meantime, please contact press@nature.com.

Please note that *Nature Structural & Molecular Biology* is a Transformative Journal (TJ). Authors may publish their research with us through the traditional subscription access route or make their paper immediately open access through payment of an article-processing charge (APC). Authors will not be required to make a final decision about access to their article until it has been accepted. <https://www.springernature.com/gp/open-research/transformative-journals> Find out more about Transformative Journals

Authors may need to take specific actions to achieve <https://www.springernature.com/gp/open-research/funding/policy-compliance-faqs> compliance with funder and institutional open access mandates. If your research is supported by a funder that requires immediate open access (e.g. according to <https://www.springernature.com/gp/open-research/plan-s-compliance> Plan S principles) then you should select the gold OA route, and we will direct you to the compliant route where possible. For authors selecting the subscription publication route, the journal's standard licensing terms will need to be accepted, including <https://www.springernature.com/gp/open-research/policies/journal-policies> self-archiving policies. Those licensing terms will supersede any other terms that the author or any third party may assert apply to any version of the manuscript.

Sincerely,

Carolina Perdigoto, PhD
Chief Editor
Nature Structural & Molecular Biology
orcid.org/0000-0002-5783-7106
